**Utsu aftershock productivity law explained from geometric operations on the**
**permanent static stress field of mainshocks**
Arnaud Mignan*
Institute of Geophysics, Swiss Federal Institute of Technology, Zurich
*Address:* ETHZ, Institute of Geophysics, NO H66, Sonneggstrasse 5, CH-8092 Zurich
*Correspondence to:* arnaud.mignan@sed.ethz.ch
*Abstract:* The aftershock productivity law is an exponential function of the form
$K \propto \exp(\alpha M)$ with $K$ the number of aftershocks triggered by a given mainshock of
magnitude $M$ and $\alpha \approx \ln(10)$ the productivity parameter. This law remains empirical
in nature although it has also been retrieved in static stress simulations. Here, we
parameterize this law using the Solid Seismicity Postulate (SSP), the basis of a
geometrical theory of seismicity where seismicity patterns are described by
mathematical expressions obtained from geometric operations on a permanent static
stress field. We first test the SSP that relates seismicity density to a static stress step
function. We show that it yields a power exponent $q = 1.96\pm0.01$ for the power-law
spatial linear density distribution of aftershocks, once uniform noise is added to the
static stress field, in agreement with observations. We then recover the exponential
function of the productivity law with a break in scaling obtained between small and
large $M$, with $\alpha = 1.5\ln(10)$ and $\ln(10)$, respectively, in agreement with results from
previous static stress simulations. Possible biases of aftershock selection, verified to
exist in Epidemic-Type Aftershock Sequence (ETAS) simulations, may explain the
lack of break in scaling observed in seismicity catalogues. The existence of the
theoretical kink remains however to be proven. Finally, we describe how to estimate
the Solid Seismicity parameters (activation density $\delta_+$, aftershock solid envelope $r_*$
and background stress amplitude range $\Delta o_*$) for large $M$ values.

**1. Introduction**

Aftershocks, one of the most studied patterns observed in seismicity, are

characterized by three empirical laws, which are functions of time, such as the
Modified Omori law (e.g., Utsu et al., 1995), space (e.g., Richards-Dinger et al., 2010;
Moradpour et al., 2014), and mainshock magnitude (Utsu, 1970a; b; Ogata, 1988).
The present study focuses on the latter relationship, i.e., the Utsu aftershock
productivity law, which describes the total number of aftershocks $K$ produced by a
mainshock of magnitude $M$ as
$$K(M) = K_0\exp[\alpha(M - m_0)] \tag{1}$$
with $m_0$ the minimum magnitude cutoff (Utsu, 1970b; Ogata, 1988). This relationship
was originally proposed by Utsu (1970a; b) by combining two other empirical laws,
the Gutenberg-Richter relationship (Gutenberg and Richter, 1944) and Båth's law
(Båth, 1964), respectively:
$$\begin{cases} N(\geq m) = A\exp[-\beta(m - m_0)] \\ \quad N(\geq M - \Delta m_B) = 1 \end{cases} \tag{2}$$
with $N$ the average number of events above magnitude $m$, $A$ a seismic activity
constant, $\beta$ the magnitude size ratio (or $b = \beta/\ln(10)$ in base-10 logarithmic scale) and
$\Delta m_B$ the magnitude difference between the mainshock and its largest aftershock, such
that
$$K(M) = N(\geq m_0 | M) = \exp(-\beta\Delta m_B)\exp[\beta(M - m_0)] \tag{3}$$
with $K_0 = \exp(-\beta\Delta m_B)$ and $\alpha \equiv \beta$. Eq. (3) was only implicit in Utsu (1970a) and
not exploited in Utsu (1970b) where $K_0$ was fitted independently of the value taken by
Båth's parameter $\Delta m_B$. The $\alpha$-value was in turn decoupled from the $\beta$–value in later
studies (e.g., Seif et al. (2017) and references therein).
Although it seems obvious that Eq. (1) can be explained geometrically if the
volume of the aftershock zone is correlated to the mainshock surface area $S$ with
$$S(M) = 10^{M-4} = \exp[\ln(10)(M - 4)] \tag{4}$$
(Kanamori and Anderson, 1975; Yamanaka and Shimazaki, 1990; Helmstetter, 2003),
there is so far no analytical, physical expression of Eq. (1) available. Although Hainzl
et al. (2010) retrieved the exponential behavior in numerical simulations where
aftershocks were produced by the permanent static stress field of mainshocks of
different magnitudes, it remains unclear how $K_0$ and $\alpha$ relate to the underlying
physical parameters.

The aim of the present article is to describe the Utsu aftershock productivity

equation (Eq. 1) in terms of a geometrical theory of seismicity coined "Solid
Seismicity", where the Eq. (4) scaling is parameterized using the Solid Seismicity
Postulate (SSP). The SSP has already been shown to effectively explain other
empirical laws of both natural and induced seismicity from simple geometric
operations on a permanent static stress field (Mignan, 2012; 2016a). The theory is
applied here for the first time to describe aftershocks.

**2. Physical Expression of the Aftershock Productivity Law**
*2.1. Demonstration of the productivity law by geometric operations*

"Solid Seismicity", a geometrical theory of seismicity, is based on the

following Postulate (Mignan et al., 2007; Mignan, 2008, 2012; 2016a):

**Solid Seismicity Postulate (SSP):** *Seismicity can be strictly categorized*
*into three regimes of constant spatiotemporal densities $\delta$ – background*
*$\delta_0$, quiescence $\delta_-$ and activation $\delta_+$ (with $\delta_- \ll \delta_0 \ll \delta_+$) - occurring*
*respective to the static stress step function:*
$$\delta(\sigma) = \begin{cases} \delta_- & , \sigma < -\Delta o_* \\ \delta_0 & , \sigma \le |\pm\Delta o_*| \\ \delta_+ & , \sigma > \Delta o_* \end{cases} \qquad (5)$$

*with $\sigma$ the static stress [stress unit], $\Delta o_*$ the background stress amplitude*
*range [stress unit], a stress threshold value separating two seismicity*

*regimes, and δ the spatial density of events [number of events per unit of*

*volume] per seismicity regime.*


We mean by "strictly categorized" that any seismicity population is either part of the

background, quiescence or activation regime (or class), with no other regime/class

possible (i.e., a sort of hard labelling). Based on this Postulate, Mignan (2012)

demonstrated the power-law behavior of precursory seismicity in agreement with the

observed time-to-failure equation (Varnes, 1989), while Mignan (2016a)

demonstrated both the observed parabolic spatiotemporal front and the linear

relationship with injection-flow-rate of induced seismicity (Shapiro and Dinske,

2009). It remains unclear whether the SSP has a physical origin or not. If not, it would

still represent a reasonable approximation of the linear relationship between event

production and static stress field in a simple clock-change model (Hainzl et al., 2010;

Fig. 1a). For the testing of the SSP on the observed spatial distribution of aftershocks,

see section 2.2. The power of Eq. (5) is that it allows defining seismicity patterns in

terms of "solids" described by the spatial envelope $r_* = r(\sigma = \pm\Delta o_*)$ where $r$ is the

distance from the static stress source (e.g., mainshock rupture) and $r_*$ the distance $r$ at

which there is a change of regime (quiescence/background at $\sigma = -\Delta o_*$ or

background/activation at $\sigma = \Delta o_*$). The spatiotemporal rate of seismicity is then a

mathematical expression defined by the density of events δ times the volume

characterized by $r_*$ (see previous demonstrations in Mignan et al. (2007) and Mignan

(2011; 2012; 2016a) where simple algebraic expressions were obtained).

In the case of aftershocks, we define the static stress field of the mainshock by

$$\sigma(r) = -\Delta\sigma_0 \left[\left(1 - \frac{c^3}{(r+c)^3}\right)^{-1/2} - 1\right] \qquad (6)$$
with $\Delta\sigma_0 < 0$ the mainshock stress drop, $c$ the crack radius and $r$ the distance from the
crack. Eq (6) is a simplified representation of stress change from slip on a planar
surface in a homogeneous elastic medium. It takes into account both the square root
singularity at crack tip and the $1/r^3$ falloff at higher distances (Dieterich, 1994; Fig.
1b). It should be noted that this radial static stress field does not represent the
geometric complexity of Coulomb stress fields (Fig. 2a). However we are here only
interested in the general behavior of aftershocks with Eq. (6) retaining the first-order
characteristics of this field (i.e., on-fault seismicity; Fig. 2b), which corresponds to the
case where the mainshock relieves most of the regional stresses and aftershocks occur
on optimally oriented faults. It is also in agreement with observations, most
aftershocks being located on and around the mainshock fault traces in Southern
California (Fig. 2c; see section "Observations & Model Fitting"). The occasional
cases where aftershocks occur off-fault (e.g., Ross et al., 2017) can be explained by
the mainshock not relieving all of the regional stress (King et al., 1994; Fig. 2d).

For $r_* = r(\sigma = \Delta o_*)$, Eq. (6) yields the aftershock solid envelope of the form:

$$r_*(c) = \left\{ \frac{1}{\left[1-\left(1-\frac{\Delta\sigma_*}{\Delta\sigma_0}\right)^{-2}\right]^{1/3}} - 1 \right\} c = Fc, \qquad (7)$$
function of the crack radius $c$ and of the ratio between background stress amplitude
range $\Delta o_*$ and stress drop $\Delta\sigma_0$ (Fig. 1c). With $\Delta\sigma_0$ independent of earthquake size
(Kanamori and Anderson, 1975; Abercrombie and Leary, 1993) and $\Delta o_*$ assumed
constant, $r_*$ is directly proportional to $c$ with proportionality constant, or stress factor,
$F$ (Eq. 7). Geometrical constraints due to the seismogenic layer width $w_0$ then yield
$$c(M) = \begin{cases} \left(\frac{S(M)}{\pi}\right)^{1/2} & , S(M) \leq \pi w_0^2 \\ w_0 & , S(M) > \pi w_0^2 \end{cases} \qquad (8)$$
with $S$ the rupture surface area defined by Eq. (4) and $c$ becoming an effective crack
radius (Kanamori and Anderson, 1975; Fig. 1d). Note that the factor of 2 (i.e., using
$w_0$ instead of $w_0/2$) comes from the free surface effect (e.g., Kanamori and Anderson,
1975; Shaw and Scholz, 2001).
The aftershock productivity $K(M)$ is then the activation density $\delta_+$ times the
volume $V_*(M)$ of the aftershock solid. For the case in which the mainshock relieves
most of the regional stress, stresses are increased all around the rupture (King et al.,
1994), which is topologically identical to stresses increasing radially from the rupture
plane (Fig. 2a-b). It follows that the aftershock solid can be represented by a volume
of contour $r_*(M)$ from the rupture plane geometric primitive, i.e., a disk or a
rectangle, for small and large mainshocks, respectively. This is illustrated in Figure
3a-b and can be generalized by
$$V_*(M) = 2r_*(M)S(M) + \frac{\pi}{2}r_*^2(M)d \qquad (9)$$
where $d$ is the distance travelled around the geometric primitive by the geometric
centroid of the semi-circle of radius $r_*(M)$ (i.e., Pappus's Centroid Theorem), or
$$d = \begin{cases} 2\pi\left(c(M) + \frac{4}{3\pi}r_*(M)\right) & , c(M) + r_*(M) \leq \frac{w_0}{2} \\ 2w_0 & , c(M) + r_*(M) > \frac{w_0}{2} \end{cases} \qquad (10)$$
For the disk, the volume (Eq. 9) corresponds to the sum of a cylinder of radius $c(M)$
and height $2r_*(M)$ (first term) and of half a torus of major radius $c(M)$ and minus
radius $r_*(M)$ (second term). For the rectangle, the volume is the sum of a cuboid of
length $l(M)$ (i.e., rupture length), width $w_0$ and height $2r_*(M)$ (first term) and of a
cylinder of radius $r_*(M)$ and height $w_0$ (second term; see red and orange volumes,
respectively, in Figure 3a-c). Finally inserting Eqs. (7), (8) and (10) into (9), we
obtain
$$K(M) = \delta_+ \begin{cases} \left[\frac{2F}{\sqrt{\pi}} + F^2\sqrt{\pi}\left(1 + \frac{4}{3\pi}F\right)\right]S^{3/2}(M) & , S(M) \leq \left(\frac{w_0\sqrt{\pi}}{2(1+F)}\right)^2 \\ \frac{2F}{\sqrt{\pi}}S^{3/2}(M) + F^2 w_0 S(M) & \left(\frac{w_0\sqrt{\pi}}{2(1+F)}\right)^2 < S(M) \leq \pi w_0^2 \\ 2F w_0 S(M) + \pi F^2 w_0^3 & , S(M) > \pi w_0^2 \end{cases}$$

(11)

which is represented in Figure 3d. Considering the two main regimes only (small
versus large mainshocks) and inserting Eq. (4) into (11), we get
$$K(M) = \delta_+ \begin{cases} \left[\frac{2F}{\sqrt{\pi}} + F^2\sqrt{\pi}\left(1 + \frac{4}{3\pi}F\right)\right]\exp\left[\frac{3\ln(10)}{2}(M-4)\right] & , \text{small } M \\ 2F w_0 \exp[\ln(10)(M-4)] + \pi F^2 w_0^3 & , \text{large } M \end{cases}$$   (12)

which is a closed-form expression of the same form as the original Utsu productivity
law (Eq. 1). Note that $K$ and $\delta_+$ are both, implicitly, function of the selected minimum
aftershock magnitude threshold $m_0$.

Here, we predict that the α-value decreases from $3\ln(10)/2 \approx 3.45$ to $\ln(10) \approx$

2.30 when switching regime from small to large mainshocks (or from 1.5 to 1 in base-
10 logarithmic scale). It should be noted that Hainzl et al. (2010) observed the same
break in scaling in static stress transfer simulations, which corroborates our analytical
findings. Hainzl et al. (2010) simulated aftershocks using the clock-change model
where events were advanced in time by the static stress change produced by a
mainshock in a three-dimensional medium. They explained the scaling break
observed in simulation as a transition from 3D to 2D scaling regime when the
mainshock rupture dimension approached $w_0$, which is compatible with the present
demonstration. For large $M$, the scaling is fundamentally the same as in Eq. (4). Since
that relation also explains the slope of the Gutenberg-Richter law (see physical
explanation given by Kanamori and Anderson,1975), it follows that $\alpha \equiv \beta$, which is
also in agreement with the original formulation of Utsu (1970a; b; Eq. 3).

*2.2. Testing of the SSP on the aftershock spatial distribution*
The SSP predicts a step-like behavior of the aftershock spatial density for an
idealized smooth static stress field (Fig. 4a-b), which is in disagreement with real
aftershock observations. A number of studies have shown that the spatial linear
density distribution of aftershocks ρ is well represented by a power-law, expressed as
$\rho(r) \propto r^{-q}$                                                            (13)
with *r* the distance from the mainshock and *q* the power-law exponent. This parameter
ranges over $1.3 \leq q \leq 2.5$ (Felzer and Brodsky, 2006; Lipiello et al., 2009; Marsan and
Lengliné, 2010; Richards-Dinger et al., 2010; Shearer, 2012; Gu et al., 2013;
Moradpour et al., 2014; van der Elst and Shaw, 2015). Although Felzer and Brodsky
(2004) suggested a dynamic stress origin for aftershocks, their results were later on
questioned by Richards-Dinger et al. (2010). Most of the studies cited above suggest
that the *q*-value is explained from a static stress process. As for the examples of
aftershocks shown to be dynamically triggered (e.g., Fan and Shearer, 2016), they are
too few to alter the aftershock productivity law and too remote to be consistently
defined as aftershocks in cluster methods.
In a more realistic setting, the static stress field must be heterogeneous (due to
the occurrence of previous events and other potential stress perturbations). We
therefore simulate the static stress field by adding a uniform random component
bounded over $\pm \Delta o_*$ following Mignan (2011) (see also King and Bowman, 2003).
Note that any deviation above $\Delta o_*$ would be flattened to $\Delta o_*$ over time by temporal
diffusion (so-called "historical ghost static stress field" in Mignan, 2016a). Figure 4c
shows the resulting stress field and Figure 4d the predicted aftershock spatial density.
Adding uniform noise blurs the contour of the aftershock solid, switching the
aftershock spatial density from a step function (Fig. 4b) to a power-law (Fig. 4d). We
fit Eq. (13) to the simulated data using the Maximum Likelihood Estimation (MLE)
method with $r_{min} = r_*$ (Clauset et al., 2009) and find $q = 1.96\pm0.01$, in agreement with
the aftershock literature. This result alone is however insufficient to prove the validity
of the SSP.

**3. Observations & Model Fitting**
*3.1. Data*
We consider the case of Southern California and extract aftershock sequences
from the relocated earthquake catalog of Hauksson et al. (2012) defined over the
period 1981-2011, using the nearest-neighbor method (Zaliapin et al., 2008; used with
its standard parameters originally calibrated for Southern California, considering only
the first aftershock generation). Only events with magnitudes greater than $m_0 = 2.0$ are
considered (a conservative estimate following results of Tormann et al. (2014);
saturation effects immediately after the mainshock are negligible when considering
entire aftershock sequences; Helmstetter et al., 2005).

*3.2. Aftershock spatial density distribution*
Figure 5a represents the spatial linear density distribution of aftershocks $\rho(r)$
for the four largest strike-slip mainshocks in Southern California: 1987 *M*=6.6
Superstition Hills, 1992 *M*=7.3 Landers, 1999 *M*=7.1 Hector Mine, and 2010 *M*=7.2
El Mayor. The distance between mainshock and aftershocks is calculated as
$r = \sqrt{(x - x_0)^2 + (y - y_0)^2}$ with $(x, y)$ the aftershock coordinates and $(x_0, y_0)$ the
coordinates of the nearest point to the mainshock fault rupture (as depicted in Figure
2c). The dashed black lines shown in Figure 5a are visual guides to $q = 1.96$, showing
that the SSP is compatible with real aftershock observations.

Comparing Figure 5a to Figure 4d suggests that $r_*$ can be roughly estimated

from the spatial linear density plot, being the maximum distance $r$ at which the
plateau ends, here leading to $r_* \approx 1$ km. This parameter is constant for different large
$M$ values since both $w_0$ and $\Delta\sigma_0$ are constant while $\Delta\sigma_*$ is also *a priori* a constant. We
can then estimate the ratio $\Delta\sigma_*/\Delta\sigma_0$ from Eq. (7). However the result is ambiguous
due to uncertainties on the width $w_0$. For $w_0 = \{5, 10, 15\}$ km, we get $\Delta\sigma_*/\Delta\sigma_0 =\{-$
$0.54, -1.01, -1.38\}$.

As for the plateau value $\rho(r < r_*)$, it provides an estimate of the aftershock

activation density $\delta_+$ with
$$\delta_+ = \frac{\rho(M, r < r_*)}{\exp[\ln(10)(M-4)]} \qquad\qquad (14)$$
a volumetric density, i.e. the linear density $\rho$ normalized by the mainshock rupture
area (Eq. 4). Due to the fluctuations in $\rho(r < r_*)$, $\delta_+$ will be estimated from the
productivity law instead (see section 3.3) and $\rho(r < r_*)$ then estimated from Eq. (14)
(horizontal dashed colored lines), as detailed below.

It should be noted that we consider only the first-generation aftershocks to

avoid $\rho$ heterogeneities from secondary aftershock clusters occurring off-fault. An
example of such heterogeneity/anisotropy is illustrated by the Landers-Big Bear case
(Fig. 2c; dotted colored curve on Fig. 5a). Those cases are not systematic and
therefore not considered in the aftershock productivity law. However they are also
due to static stress changes (e.g., King et al., 1994) with the anisotropic effects
explainable by Solid Seismicity through the concept of "historical ghost static stress
field" (Mignan, 2016a).

*3.3. Aftershock productivity law*

247    The observed number $n$ of aftershocks of magnitude $m \geq m_0$ produced by a

248 mainshock of magnitude $M$ (for a total of $N$ mainshocks) in Southern California is

249 shown in Figures 5b (for large $M \geq 6$) and 6a (for the full range $M \geq m_0$). We fit Eq.

250 (1) to the data using the MLE method with the log-likelihood function

251 $LL(\theta; X = \{n_i; i = 1, ..., N\}) = \sum_{i=1}^{N}[n_i\ln[K_i(\theta)] - K_i(\theta) - \ln(n_i!)]$   (15)

252 for a Poisson process, representing the stochasticity of the count $K$ of aftershocks

253 produced by a mainshock at any given time. Inserting Eq. (1) in Eq. (15) yields

254 $LL(\theta = \{K_0, \alpha\}; X) = \ln(K_0)\sum_{i=1}^{N} n_i + \alpha \sum_{i=1}^{N}[n_i(M_i - m_0)] - K_0 \sum_{i=1}^{N} \exp[\alpha(M_i -$

255 $m_0)] - \sum_{i=1}^{N} \ln(n_i!)$              (16)

256 (note that the last term can be set to 0 during $LL$ maximization). For Southern

257 California, we obtain $\alpha_{MLE} = 2.32$ (1.01 in $\log_{10}$ scale) and $K_0 = 0.025$ when

258 considering large $M \geq 6$ mainshocks only to avoid the issues of scaling break and data

259 dispersion at lower magnitudes. This result, represented by the black solid line on

260 Figure 5b, is in agreement with previous studies in the same region (e.g., Helmstetter,

261 2003; Helmstetter et al., 2005; Zaliapin and Ben-Zion, 2013; Seif et al., 2017) and

262 with $\alpha = \ln(10) \approx 2.30$ predicted for large mainshocks in Solid Seismicity (Eq. 12).

263 Moreover we find a bulk $\beta_{MLE} = 2.34$ (1.02 in $\log_{10}$ scale) (Aki, 1965), in agreement

264 with $\alpha \equiv \beta$.

265    Let us now rewrite the Solid Seismicity aftershock productivity law (Eq. 12)

266 by only considering the large $M$ case and injecting $r_* = Fw_0$ (by combining Eqs. 7-8).

267 We get

268 $K(M > M_{break}) = \delta_+\{2r_*\exp[ln(10)(M - 4)] + \pi r_*^2 w_0\}$   (17)

269 The role of $w_0$ is illustrated in Figure 5b for different values (dashed and dotted

270 curves) and shown to be insignificant for large $M$ values. Therefore Eq. (17) can be

271 approximated to

$$K(M > M_{break}) \approx 2\delta_+ r_* \exp[ln(10)(M - 4)] \qquad (18)$$
By analogy with Eq. (1), we get
$$\delta_+ = \frac{K_0 \exp[\ln(10)(4-m_0)]}{2r_*} \qquad (19)$$
With $r_* \approx 1$ km estimated from $\rho(r)$ (section 3.2) and $K_0 = 0.025$, we obtain $\delta_+ = 1.23$
events/km$^3$ for $m_0 = 2$. We then get back the plateau $\rho(r < r_*)$ for different $M$ values
from Eq. (14), as shown in Figure 5a (horizontal dashed colored lines). Although
based on limited data, this result suggests that the activation parameter $\delta_+$ is constant
(at least for large $M$) in Southern California. Note that if $\rho(r < r_*)$ was well
constrained, it could have been estimated jointly with $r_*$ from Figure 5a to predict the
aftershock productivity law of Figure 5b without further fitting required (hence
removing $K_0$ from the equation, $K_0$ having no physical meaning in Solid Seismicity).

**4. Role of aftershock selection on productivity scaling-break**
We tested the following piecewise model to identify any break in scaling at
smaller $M$, as predicted by Eq. (12):
$$K(M) = \begin{cases} K_0 \frac{\exp[\ln(10)(M_{break}-m_0)]}{\exp\left[\frac{3}{2}\ln(10)(M_{break}-m_0)\right]} \exp\left[\frac{3}{2}\ln(10)(M - m_0)\right] & , M \leq M_{break} \\ K_0 \exp[\ln(10)(M - m_0)] & , M > M_{break} \end{cases}$$

(20)

but with the best MLE result obtained for $M_{break} = m_0$, suggesting no break in scaling
in the aftershock productivity data, as observed in Figure 6a. Final parameter
estimates are $\alpha_{MLE} = 1.95$ (0.85 in $\log_{10}$ scale) and $K_0 = 0.141$ for the full mainshock
magnitude range $M \geq m_0$ (dotted line), subject to high scattering at low $M$ values.
We now identify whether the lack of break in scaling in aftershock
productivity observed in earthquake catalogues could be an artefact related to the
aftershock selection method. We run Epidemic-Type Aftershock Sequence (ETAS)
simulations (Ogata, 1988; Ogata and Zhuang, 2006), with the seismicity rate
$$\begin{cases} \lambda(t,x,y) = \mu(t,x,y) + \sum_{i:t_j<t} K(M_i) f(t-t_i) g(x-x_i, y-y_i|M_i) \\ \qquad\qquad f(t) = c^{p-1}(p-1)(t+c)^{-p} \\ g(x,y|M) = \frac{1}{\pi} \left( d e^{\gamma(M-m_0)} \right)^{q-1} \left( x^2 + y^2 + d e^{\gamma(M-m_0)} \right)^{-q} (q-1) \end{cases} \qquad (21)$$

Aftershock sequences are defined by power laws, both in time and space (for an
alternative temporal function, see Mignan (2015; 2016b); the spatial power-law
distribution is in agreement with Solid Seismicity in the case of a heterogeneous static
stress field – see section 2.2). $\mu$ is the Southern California background seismicity, as
defined by the nearest-neighbor method (with same $t$, $x$, $y$ and $m$). We fix the ETAS
parameters to $\theta = \{c = 0.011$ day, $p = 1.08$, $d = 0.0019$ km$^2$, $q = 1.47$, $\gamma = 2.01$, $\beta =$
2.29, $K_0 = 0.08\}$, following the fitting results of Seif et al. (2017) for the Southern
California relocated catalog and $m_0 = 2$ (see their Table 1). However, we define the
productivity function $K(M)$ from Eq. (20) with $M_{break} = 5$. Examples of ETAS
simulations are shown in Figure 6b for comparison with the observed Southern
California time series. Figure 6c allows us to verify that the simulated aftershock
productivity is kinked at $M_{break}$, as defined by Eq. (20).

We then select aftershocks from the ETAS simulations with the nearest-

neighbor method. Figure 4d represents the estimated aftershock productivity, which
has lost the break in scaling originally implemented in the simulations (with an
underestimated $\alpha_{MLE} = 2.07$ as observed in the real case for $M \geq m_0$). Note that a
similar result is obtained when using a windowing method (Gardner and Knopoff,
1974). This demonstrates that the theoretical break in scaling predicted in the
aftershock productivity law can be lost in observations due to an aftershock selection
bias, all declustering techniques assuming continuity over the entire magnitude range.
While such a bias is possible, it yet does not prove that the break in scaling exists. The
fact that a similar break in scaling was obtained in independent Coulomb stress
simulations (Hainzl et al., 2010) however provides high confidence in our results.

One other possible explanation for lack of scaling break is that our

demonstration assumes moment magnitudes while the Southern California catalogue
is in local magnitudes. Deichmann (2017) demonstrated that while $M_L \propto M_w$ at large
M, $M_L \propto 1.5 M_w$ at smaller $M$ values. This could in theory cancel the kink in real data.
However the scaling break predicted by Deichmann (2017) occurs at several
magnitude units below the geometric scaling break expected by Solid Seismicity,
invalidating this second option for mid-range magnitudes $M$.

**5. Conclusions**

In the present study, a closed-form expression defined from geometric and

static stress parameters was proposed (Eq. 12) to describe the empirical Utsu
aftershock productivity law (Eq. 1). This demonstration is similar to the previous ones
made by the author to explain precursory accelerating seismicity and induced
seismicity (Mignan, 2012; 2016b), In all these demonstrations, the main physical
parameters remain the same, i.e. the activation density $\delta_+$ (also $\delta_-$ and $\delta_0$), the
background stress amplitude range $\Delta o_*$, and the solid envelope $r_*$ which describes the
geometry of the "seismicity solid" (Fig. 3a-b). Further studies will be needed to
evaluate whether the $\delta_+$ and $\Delta o_*$ parameters are universal or region-specific and if the
same values apply to different types of seismicity at a same location.

Although the Solid Seismicity Postulate (SSP) (Eq. 5) remains to be proven, it

is so far a rather convenient and pragmatic assumption to determine the physical
parameters that play a first-order role in the behavior of seismicity. The similarity of
the SSP-simulated and observed values of the power-law exponent $q$ of the aftershock
spatial density distribution shows that the SSP is consistent with large aftershock
observations once uniform noise is added to the stress field (Figs. 4d-5a). The impact
of other types of noise on $q$ has yet to be investigated. The SSP is also complementary
to the more common simulations of static stress loading (King and Bowman, 2003)
and static stress triggering (Hainzl et al., 2010).

Analytic geometry, providing both a visual representation and an analytical

expression of the problem at hand (Fig. 3), represents a new approach to try to better
understand the behavior of seismicity. Its current limitation in the case of aftershock
analysis consists in assuming that the static stress field is radial and described by Eq.
(6) (e.g., Dieterich, 1994), which is likely only valid for mainshocks relieving most of
the regional stresses and with aftershocks occurring on optimally oriented faults (King
et al., 1994). More complex, second-order, stress behaviors might explain part of the
scattering observed around Eq. (1) (Fig. 6a), such as overpressure due to trapped high-
pressure gas for example (Miller et al., 2004 – see also Mignan (2016a) for an
overpressure field due to fluid injection). Other $\sigma(r)$ formulations could be tested in
the future, the only constraint on generating so-called seismicity solids being the use
of the postulated static stress step function of Eq. (5) (i.e., the Solid Seismicity
Postulate, SSP).

Finally, the disappearance of the predicted scaling break in the aftershock

productivity law once declustering is applied (Fig. 6) indicates that more work is
required in that domain. Only a declustering technique that does not dictate a constant
scaling at all $M$ will be able to identify rather a scaling break really exists or not.

*Acknowledgments:* I thank N. Wetzler and two anonymous reviewers, as well as
editor Ilya Zaliapin, for their valuable comments.

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

**499 Figures**

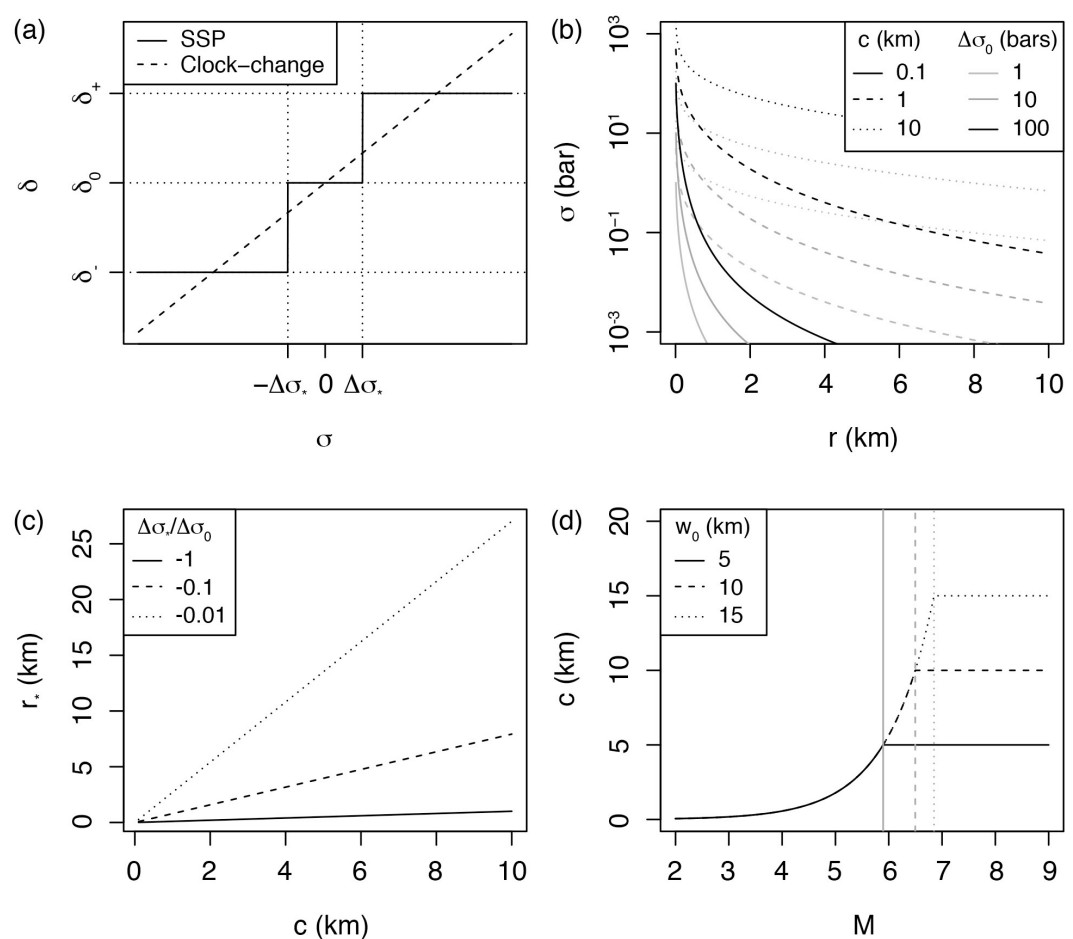

**Figure 1.** Definition of the aftershock solid envelope in a permanent static stress field:
(a) Event density stress step-function $\delta(\sigma)$ (Eq. 5) of the Solid Seismicity Postulate
(SSP) in comparison to the linear clock-change model; (b) Static stress $\sigma$ versus
distance $r$ for different effective crack radii $c$ and rupture stress drops $\Delta\sigma_0$ (Eq. 6); (c)
Linear relationship between effective crack radius $c$ and aftershock solid envelope
radius $r_*$ for different $\Delta\sigma_*/\Delta\sigma_0$ ratios (Eq. 7); (d) Relationship between mainshock
magnitude $M$ and effective crack radius $c$ for different seismogenic widths $w_0$ (Eq. 8).

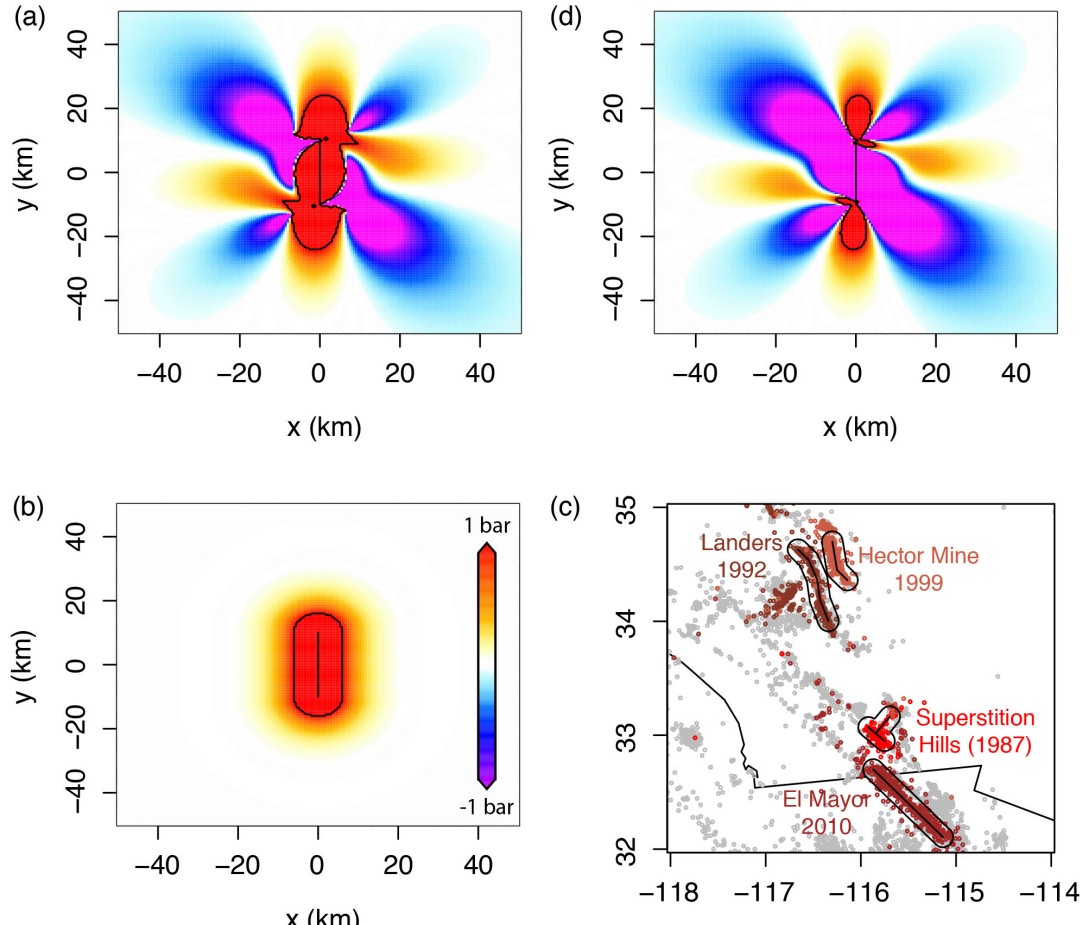


**Figure 2.** Possible static stress fields and inferred aftershock spatial distribution: (a)
Right-lateral Coulomb stress field for optimally oriented faults, where the mainshock
relieves all of the regional stresses $\sigma_r = 10$ bar, with $\Delta\sigma_0 \approx -Gs/L \approx$ - 10 bar ($G =$
$3.3.10^5$ bar the shear modulus, $s = 0.6$ m the slip, $L = 20$ km the fault length, and $w =$
10 km the fault width); (b) Radial static stress field computed from Eq. (6) with $\Delta\sigma_0 =$
-10 bar and $c = \sqrt{(Lw)/\pi}$ for consistency with (a); (c) Aftershock distribution of the
largest strike-slip events in the Southern California relocated catalog, identified here
as all events occurring within one day of the mainshock (see Data section 3.1); (d)
Right-lateral Coulomb stress field for optimally oriented faults, where the mainshock
relieves only a fraction of the regional stresses $\sigma_r$ = 100 bar with $\Delta\sigma_0$ = -10 bar (same
rupture as in (a)) – The black contour represents 1 bar in (a), (b) and (d), and a 10 km
distance from rupture in (c). Coulomb stress fields of (a) and (d) were computed using
the Coulomb 3 software (Lin and Stein, 2004; Toda et al., 2005).

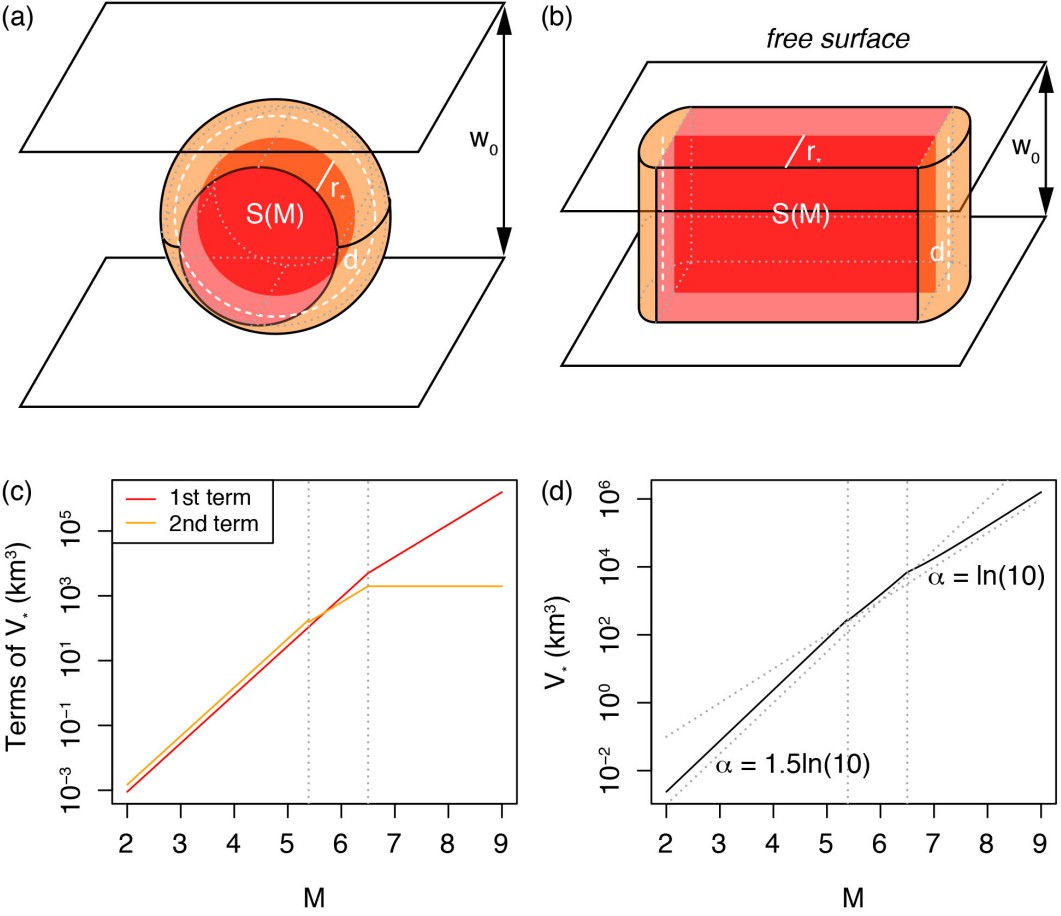


**Figure 3.** Geometric origin of the aftershock productivity law: (a) Sketch of the
aftershock solid for a small mainshock rupture represented by a disk; (b) Sketch of the
aftershock solid for a large mainshock rupture represented by a rectangle; (c) Relative
role of the two terms of Eq. (9), here with $w_0$ = 10 km and $\frac{\Delta\sigma_*}{\Delta\sigma_0}$ = -0.1 (to first estimate
c and $r_*$ from Eqs. 8 and 7, respectively); (d) Aftershock productivity law (normalized
by $\delta_+$) predicted by Solid Seismicity (Eq. 11). This relationship is of the same form as
the Utsu productivity law (Eq. 1) for large $M$ (see text for an explanation of the lack
of break in scaling in Eq. 1 for small $M$). Dotted vertical lines represent $M$ for
$c(M) + r_*(M) = \frac{w_0}{2}$ and $S(M) = \pi w_0^2$, respectively.

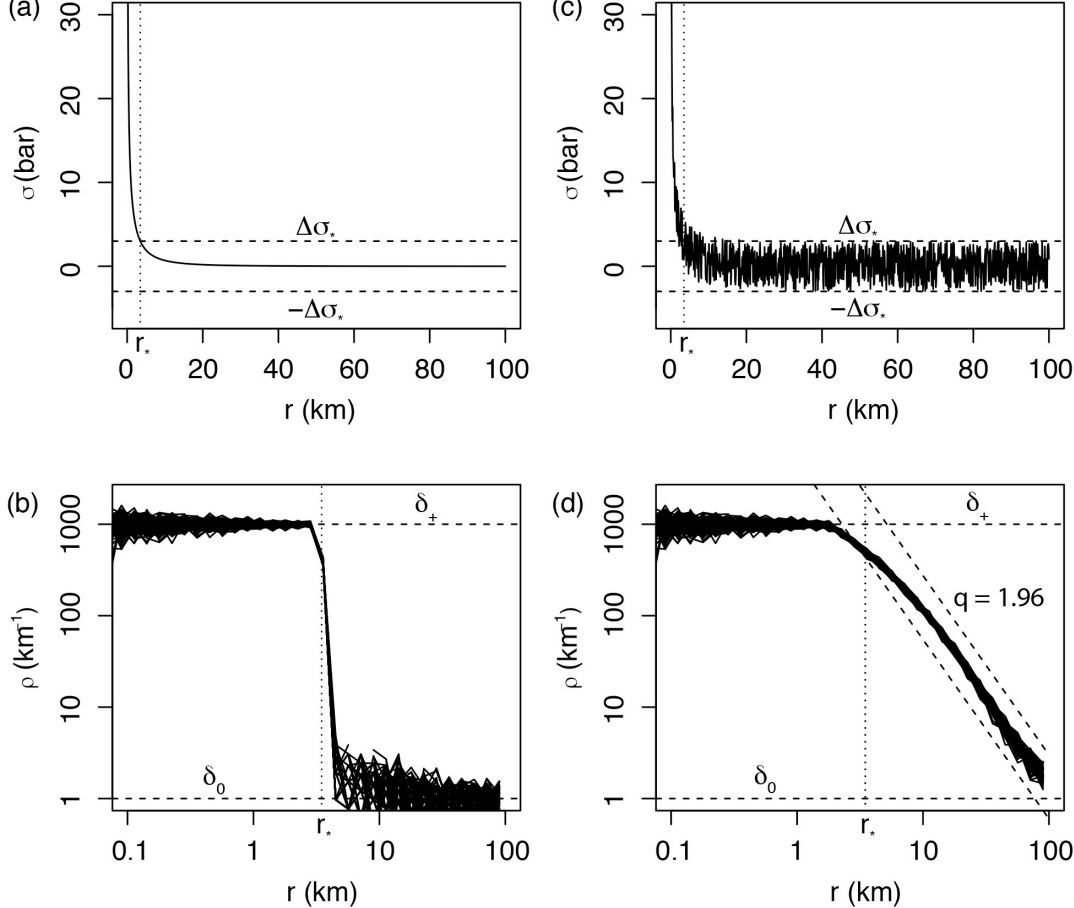


**Figure 4.** Spatial distribution of aftershocks following the SSP. (a) Smooth static
stress field as a function of distance $r$ from the mainshock, with $\Delta\sigma_0 = -10$ bar and $c =$
10 km (Eq. 6); (b) Step-like aftershock spatial linear density $\rho(r)$ with $\delta_+ = 1000$
events per km, $\delta_0 = 1$ event per km and $\Delta\sigma_* = -0.3\Delta\sigma_0$ (*ad-hoc* ratio yielding $r_* = 3.5$
km; Eq. (7) – event distances sampled from the $\delta(r)$ distribution, repeated 100 times).
Such distribution is not observed in Nature; (c) Same as (a) but with random uniform
noise representative of spatial heterogeneities added to the regional stress field; (d)
Power-law-like aftershock spatial linear density $\rho(r)$ with power exponent MLE
estimate $q = 1.96$, representative of real aftershock observations (see Fig. 5a), due to
the addition of uniform noise to the static stress field.

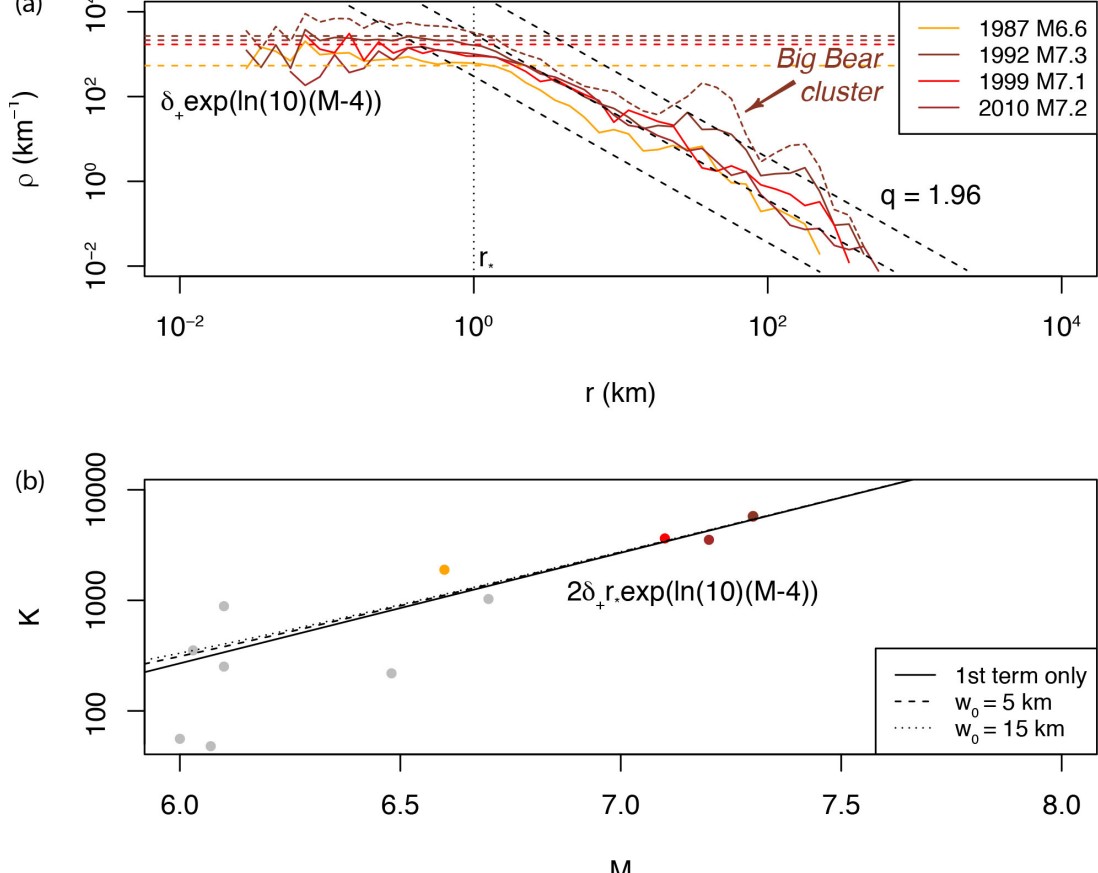


**Figure 5.** Estimating the Solid Seismicity parameters from the spatial distribution of
aftershocks: (a) Spatial linear density distribution $\rho(r)$ of aftershocks for the four
largest strike-slip mainshocks in Southern California (with first-generation
aftershocks only; the density distribution comprising all aftershocks generated by the
Landers mainshock is represented by the dotted curve to illustrate the type of spatial
heterogeneity, such as the Big Bear cluster, not considered in the present study – see
also Fig. 2c). The Solid Seismicity parameters $r_* = 1$ km and $\delta_+(m_0 = 2) = 1.23$
events/km$^3$ can be retrieved from the observed plateau $\rho(r < r_*)$, in agreement with the
SSP (see Fig. 4d). Note that the spatial power-law decay at high $r$ is similar to the one
expected by the SSP in the case of a static stress field with additive uniform noise
(expected $q = 1.96$ represented by the dashed black lines); (b) Aftershock productivity
$K$ for $M > 6$. The curves represent the productivity law as defined by Solid Seismicity
(Eq. 17) for different $w_0$ values (first term only corresponds to $w_0 = 0$; Eq. 18).

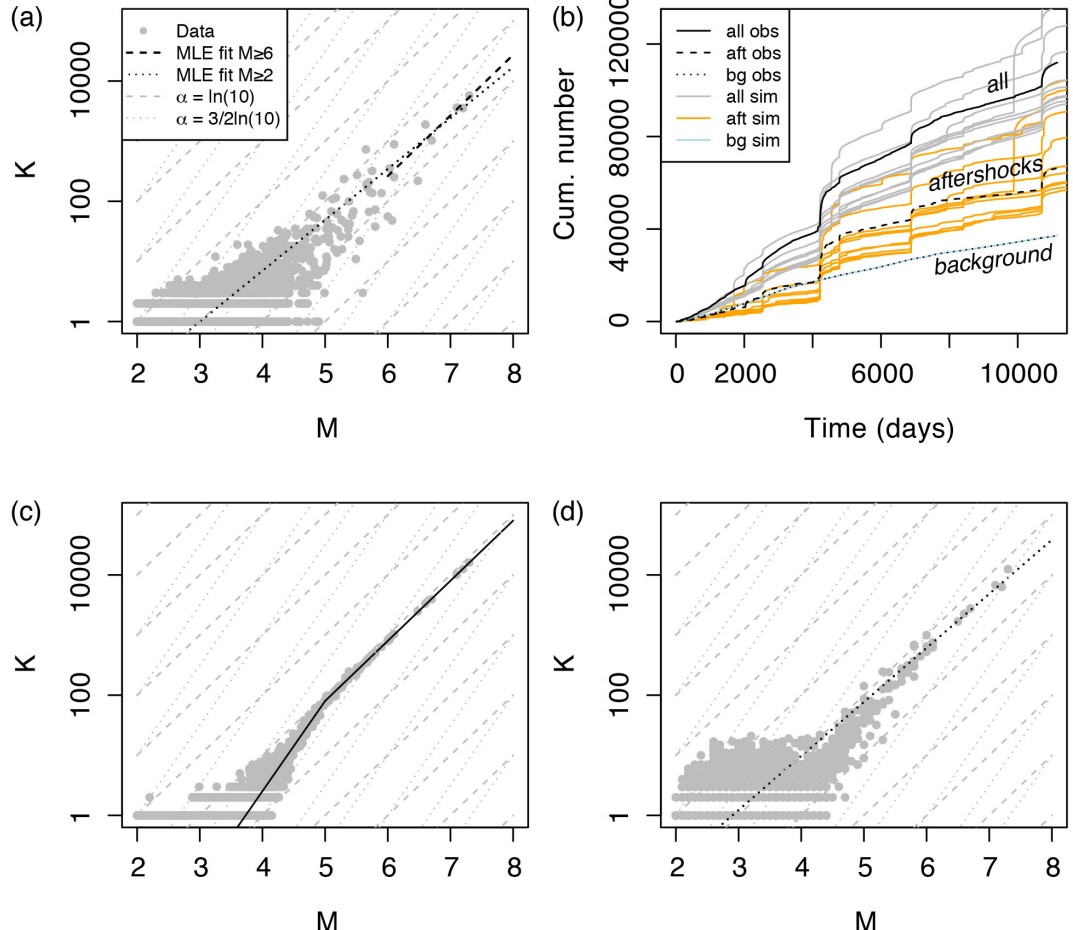


**Figure 6.** Aftershock productivity defined as the number of aftershocks $K(m_0 = 2)$ per
mainshock of magnitude $M$: (a) Observed aftershock productivity in Southern
California with aftershocks selected using the nearest-neighbor method; (b)
Seismicity time series with distinction made between background events and
aftershocks, observed ("obs", in black) and ETAS-simulated ("sim", colored); (c)
True simulated aftershock productivity with kink, defined from Eq. (20); (d)
Retrieved simulated aftershock productivity with aftershocks selected using the
nearest-neighbor method - Data points in (a), (c) and (d) are represented by grey dots;
the model MLE fits are represented by the dashed and dotted black lines for $M \geq 6$
and $M \geq m_0$, respectively; dashed and dotted grey lines are visual guides to $\alpha =$
$3/2\ln(10)$ and $\ln(10)$, respectively.