# Peer review of "Utsu aftershock productivity law explained from geometric operations on the permanent static stress field of mainshocks"

_Nonlinear Processes in Geophysics, 2017_

## Short Comment (SC1) · 23 Aug 2017

The manuscript examines the empirical relationship of the power law aftershock productivity law. The author introduces (not only in this study) the Solid Seismicity Postulate (SSP) to predict the first order mainshock's geometrical static stress perturbation on the crustal ambient stress. The model defines two basic ruptures with respect to the free surface predicting a magnitude dependent deficiency when the rupture hits the surface. Using this physical model he explains the empirical observation. The manuscript is written well and figures are useful. I have two general comments: 1) The role of dynamic triggering. In general aftershock productivity is a product of the

static and dynamic perturbation superimposed on the regional seismic susceptibility, or faults state [Dieterich, 1994]. Many examples for both dynamic triggering and Coulomb stress explain aftershocks occurrence. Due to the rapid decay of the static stress field, cases of "pure" dynamic triggering are common beyond several fault dimensions rom the mainshock [e.g. Fan and Shearer, 2016]. In the periphery of the fault, the Coulomb stress field and dynamic stress field overlap with a similar fashion, and it is unclear how they interact. My main concern is that the author does not discuss the contribution of dynamic triggering to the aftershock productivity. Does the fact that the predicted "kink" in the aftershock productivity from the geometrical interaction with the surface is due to enhancement of the dynamic triggering? 2) The geometry of the SSP. The first order shape of the SSP is not obvious to me. The geometry of the induced area is predicting a volumetric increase in static stress changes along the rupture area (red in Figure. 3). The rupture of faulted area is the expression of the coseismic slip responding to the elastic rebound. This predicts different degrees of relaxation with respect to the mainshock magnitude and the occurrence of the event in the seismic cycle. In the case of "complete" stress drop the rupture area is predicted to present spatial deficiency in productivity and some variations in the field with respect to the fault complexity. Several papers demonstrate the deficiency in aftershocks at the asperity with the majority of the seismicity focused on the periphery of the fault [Hasegawa et al., 2012; van der Elst and Shaw, 2015; Ross et al., 2017] represented by the orange volume in Figure. 3. My concern is that this model (SSP) is too simplified and does not incorporate basic modern observations.

3) Further clarification regarding the time and spatial windows used for aftershock counting for the case of Southern California is needed

Dieterich, J. H. (1994), A constitutive law for rate of earthquake production and its application to earthquake clustering, J. Geophys. Res., 99(B2), 2601–2618, doi:10.1029/93JB02581. van der Elst, N. J., and B. E. Shaw (2015), Larger aftershocks happen farther away: Nonseparability of magnitude and spatial distributions of

aftershocks, Geophys. Res. Lett., 42(14), 5771–5778, doi:10.1002/2015GL064734. Fan, W., and P. M. Shearer (2016), Local near instantaneously dynamically triggered aftershocks of large earthquakes, Science (80-. )., 353(6304), 1133–1136, doi:10.1126/science.aag0013. Hasegawa, A., K. Yoshida, Y. Asano, T. Okada, T. Iinuma, and Y. Ito (2012), Change in stress field after the 2011 great Tohoku-Oki earthquake, Earth Planet. Sci. Lett., 355–356, 231–243, doi:10.1016/j.epsl.2012.08.042. Ross, Z. E., H. Kanamori, and E. Hauksson (2017), Anomalously large complete stress drop during the 2016 M w 5.2 Borrego Springs earthquake inferred by waveform modeling and near-source aftershock deficit, Geophys. Res. Lett., 1–8, doi:10.1002/2017GL073338.

---

## Referee Comment (RC1) · Anonymous Referee #1 · 30 Oct 2017

The MS presented by Dr. Mignan intends to provide the background of the aftershock productivity law where the number of aftershock is proportional to the exponential of the magnitude ($M$) of a mainshock. On the basis of "Solid Seismicity Postulate"(SSP), the author derives the formula of the expected number of aftershocks as a function of $M$ which agrees with the productivity law originally suggested by Utsu [1970]. The derived formula has a break in the log-linear relationship between the aftershock productivity and $M$ whereas the break is not found through the analysis of real aftershock data. The author suggests that this inconsistency is caused by an aftershock selection bias with a numerical simulation.

[Figure]

I have two major concerns on this MS as shown below.

a) I do not understand well what new significant results are in this MS. In Hainzl et al [2010, JGR], the aftershock productivity law has already been reproduced with a numerical simulation. The simulation is based on the "clock-advanced" model, which is a simple but realistic physical assumption.

By contrast, SSP is too simple, and because of this simplification its physical background seems obscure and unrealistic. Furthermore, the postulate has not been supported by real data (In some of the author's previous papers, seismicity model derived from SSP has been applied to real seismicity data. Note that, however, only temporal patterns of seismic activity are analyzed. To validate SSP where we have only three seismicity levels in space, it is indispensable to reproduce spatial patterns of real earthquakes.). This MS does not show any convincing motivation to explain the productivity law with such an unsupported postulate.

I understand that sometimes it is important to introduce a (too) simple model/assumption for explaining an empirical law. However, it is also important to provide some new and meaningful perspective as a result of the introduction. The results shown in this MS do not go beyond the results of Hainzl et al. [2010], and therefore the introduction of SSP is unproductive.

b) In the end of Section 3, the author suggests the break in scaling in the aftershock productivity data (Eq.(16)). However, as a result of the analysis of the real aftershock data, no break is found (L.182-183). To explain the result of "no break", in Section 4 the numerical simulation with the ETAS model was conducted. Then, the author ascribes this result to the "aftershock selection bias" (L.206-207) in the numerical simulation.

The author's conclusion is one possibility, but it is also possible that Eq.(16) is incorrect; the numerical simulation shown in Section 4 is inconclusive, and I do not understand what is the meaning of showing such a vague consequence. The application of an aftershock selection approach having a serious problem (the bias in

this case) itself is inappropriate. Why does the author use any other approach which does not contain such a problem?

In other words, only a negative possibility for the postulate is shown and no positive support is not given in the present form of this MS. To my opinion, this is another major drawback of this MS.

Some further comments:

1) Introduction of the Zero-Inflated Poisson (ZIP) distribution

The reason of the introduction of the ZIP distribution is described in L.165-166 ("this approach ... zero aftershock"), but this explanation seems insufficient. Behind the ZIP distribution, we have the following assumption. We have two possibilities: the first is that the number of events follows a Poisson distribution, and the second is that it is deterministically equal to zero. One of these two possibilities is chosen through the Bernoulli distribution.

As far as I know, a physical (seismological) process corresponding to the Bernoulli distribution is unclear in generating earthquakes. If the author persists in introducing the ZIP distribution, explain what is the physical process.

2) The simulation shown in Section 4

This simulation is based on the ETAS model, and this violates the self-consistency of this MS. As seen in $g(x, y|M)$ of Eq.(17), the spatial density of aftershocks gradually decays with the distance from a parent event. This property completely disagree with SSP (see Eq.(5)). For the self-consistency, the simulated spatial (and temporal) pattern of earthquakes should be generated on the basis of SSP.

3) $\alpha = 2.04$ (L.197)

I do not understand how the author incorporated this information (value) into Eq.(16).
* * *
2017-38, 2017.

---

## Author Comment (AC1) · 9 Nov 2017

Dear reviewer,

Thank you for your comments on the discussion paper by Mignan (2017). Below is my two-part answer to (1) show that the Solid Seismicity Postulate is supported by seismicity data and (2) discuss in more detail the mismatch between theoretical scaling break and lack of break in real data. A third section answers to your other comments.

1 Support of the Solid Seismicity Postulate (SSP) by aftershock data

The SSP should indeed be verified to be consistent with the spatial distribution of seismicity data. I first clarify that the step-like function of event density in space is only expected for the case of an idealised smooth static stress field. I now compare this case (Fig. X1a-b) with the case of a stress field with uniform noise (Fig. X1c-d). While the ideal case is used to develop analytical solutions, a heterogeneous stress field described by additive uniform noise was already used in past studies to simulate non-stationary background events (King and Bowman, 2003; Mignan et al., 2007; Mignan, 2011). Addition of such noise blurs the "aftershock solid", which reflects in the aftershock spatial density distribution, switching from a step function to a power-law of the form rho(r) proportional to r^(-q), with rho the linear spatial density and exponent q = 1.7. Figure X1 will be inserted in the revised manuscript as a new Figure 2 (with a new paragraph inserted line 111).

As shown in Figure X2 (new Figure 5 in the revised manuscript), the power law exponent obtained from the SSP with noisy static stress field matches the power law exponent found in Southern California. In the literature, $1.3 < q < 2.5$ centred around $q = 1.4$-$1.8$ was found for California (Felzer and Brodsky, 2006; Lipiello et al., 2009; Marsan and Lengliné, 2010; Richards-Dinger et al., 2010; Shearer, 2012; Gu et al., 2013; Moradpour et al., 2014; van der Elst and Shaw, 2015). This demonstrates that the SSP is not "too simple" or "unrealistic". Comparison of Figure X1d with Figure X2a shows that "the spatial patterns of real earthquakes are reproduced" by the SSP (i.e., the power-law behaviour) with a realistic q-value (without any tuning required). A short review of past studies on the spatial distribution of aftershocks and a discussion of Figure X2 will be added line 184 in the revised manuscript (end of section 3 on "Observations & Model Fitting").

This work goes beyond the results of Hainzl et al. (2010) since an analytical formulation is explicit while the physical driver of a simulation output is implicit and potentially ambiguous. In the King and Bowman (2003) study for example, a power-law behaviour of precursory seismicity emerged from their static stress simulations. However the result was ambiguous. It was not clear if the behaviour emerged from the stress field

geometry, implemented Gutenberg-Richter power-law, or else. It led to the first study on Solid Seismicity, which demonstrated that the power-law time-to-failure equation derived from the geometry of the stress field (Mignan et al., 2007). While such ambiguity may not be present in the simulations of Hainzl et al. (2010), we are still left wondering which parameters are critical to the emergence of the Utsu productivity law, i.e., "it remains unclear how K0 and $\alpha$ relate to the underlying physical parameters" (line 50).

Here are two "new and meaningful perspectives as a result of the introduction" of the SSP: (i) It is first of importance to demonstrate that the Solid Seismicity theory can explain the aftershock productivity law, since it already explains both tectonic foreshocks (Mignan, 2012) and induced seismicity (Mignan, 2016). If such physical framework can explain the main seismicity patterns observed in Nature, it becomes a potential candidate for a unified theory of seismicity. (ii) Figure X2 goes farer into the Solid Seismicity analysis, showing how to estimate its main parameters (intermediary parameter r_*, main parameters $\delta$_+ and $\Delta\sigma$_*). We first note that the q = 1.7 theoretical estimate (SSP + uniform noise) is compatible with observations (Fig. X2a). I here focus on the largest mainshocks to avoid the scattering and scaling break issues at small M. On the same plot, we can roughly estimate r_* = 1 km (maximum r at which the rho plateau breaks – in analogy with Fig. X1d). It is constant for any large M (> Mbreak) since the stress drop is a constant, c = w_0 is a constant, and $\Delta\sigma$_* is also a priori a constant (one of the 2 main parameters of the Solid Seismicity approach; Eq. 7). Now let us calculate $\delta$_+ from the commonly used parameter K_0. We first note from Eq. (11) that the second term is negligible for large M, yielding

K(M>M_break )$\approx$2$\delta$_+(m_0) r_* exp[ln(10)(M-4)] (X1)

Rearranging m_0 and M-4 and comparing to the original Utsu Eq. (1), we get

$\delta$_+ (m_0)=K_0 exp[ln(10)(4-m_0)]/(2r_*) (X2)

With $\alpha$ = ln(10) fixed and K_0 estimated from the MLE for M > 6, we get K_0 = 0.027 and thus $\delta$_+(m_0 = 2) = 1.35 events/km^3 (fit represented in Fig. X2b). If correct, the

linear density below r_* (plateau) for any given large M should be

rho(r<r_*,M)=$\delta$_+ + exp[ln(10)(M-4)] (X3)

which is represented on Fig. X2a and matches the data (Eq. (X3) simply calculates the linear density of events rho from the volumetric density of events $\delta$_+). This suggests that $\delta$_+ is also constant, at least for the four largest strike-slip mainshocks in Southern California. One could have also estimated $\delta$_+ directly from rho(r) (as done for r_*) to directly derive the aftershock productivity law of Southern California with Eq. (X1). This shows the direct link between aftershock productivity and aftershock spatial distribution (or geometry). As for the parameter $\Delta\sigma$_*, its estimation remains ambiguous as it depends on the seismogenic width w_0. We get the ratio $\Delta\sigma$_*/$\Delta\sigma$_0 = {-0.5, -1.0, -1.4} for w_0 = {5, 10, 15} km, respectively (Eq. 7). This analysis as well as Figure X2 (new Figure 5) will be inserted at the end of section 3 in the revised manuscript. This of course remains a preliminary analysis. However I hope that additional analyses of aftershocks, foreshocks as well as induced seismicity in different regions will provide useful information as to the distribution of the $\Delta\sigma$_* and $\delta$_+ parameters. Are they universal? Is a same regional value applicable to all types of seismicity? Are there any correlations? Those are important questions I wish to answer in the near future. To do so, the theoretical framework must first be conveyed for each class of seismicity pattern.

2 Theoretical scaling break & mismatch with seismicity data

The discussion paper already indicates that: "Possible biases of aftershock selection may explain the lack of break" (lines 18-20, abstract) and "while such a bias is possible, it yet does not prove that the break in scaling exists" (line 208) – This clearly suggests that it is only one possible option. It is indeed a weak argument (since based on a negative result) but it is so far the best one available (all existing declustering techniques assuming no break in magnitude). "It is also possible that Eq. (16) is incorrect", true, but so would the clock-advance model in such premise, which the reviewer describes

as "a simple but realistic physical assumption". No explanation for the lack of break in real data was given in Hainzl et al. (2010). The present paper provides one possible explanation. Any criticism on the scaling break mismatch shall apply the same way to the present study and the published one of Hainzl et al. (2010). An alternative view is that both studies found the same scaling break, hence supporting this result as characteristic of the static stress process.

Following on the new results presented in Figure X1d, the explanation of lack of break due to aftershock selection bias becomes a more realistic one. It is NOT "a vague consequence" since any study of the aftershock productivity law is based on the use of such a declustering method. The ETAS simulation does NOT "violate the self-consistency of this MS" either since the power-law spatial distribution is now shown to be verified by the SSP. The theoretical value q = 1.7 is very close to the value I already used in the ETAS simulations (q = 1.47) and observed here for the largest strike-slip mainshocks (Fig. X2a). Since the aftershock selection bias is only one option, another alternative will be discussed: The proposed productivity equation assumes moment magnitude while the earthquake catalogue is in local magnitude. Deichmann (2017) recently demonstrated that while $M\_L$ is proportional to $M\_w$ at large M, $M\_L$ is proportional to $1.5M\_w$ at small M. This would cancel the kink observed in the real data. However the scaling break predicted by Deichmann (2017) occurs at several magnitude units below the geometric one expected by static stress.

3 Other aspects

On the introduction of the Zero-Inflated Poisson (ZIP) distribution: Explaining the distribution of earthquakes, from the static stress process to their occurrence on a fractal network of faults remains out of the scope of the present study. Since the ZIP does not lead to significant changes in the $\alpha$-value and since section 3 will be completed with an analysis of the spatial distribution of aftershocks (Fig. X2), the ZIP part will be deleted from the revised manuscript.

On $\alpha$ = 2.04 (line 197): This is the maximum likelihood estimate of $\alpha$ obtained for Southern California in the present study (see line 164). $\alpha$ is thus constrained from large magnitude data (Fig. 4a) and the simulated break at lower magnitudes is estimated from the theoretical value 3/2 $\alpha$.

Figures

Figure X1. Spatial distribution of aftershocks following the SSP. (a) Smooth static stress field as a function of distance r from the mainshock, with $\Delta\sigma\_0$ = -10 bar and c = 10 km (Eq. 6); (b) Step-like aftershock spatial linear density rho with $\delta\_+$ = 1000 events per km, $\delta\_0$ = 1 event per km and $\Delta\sigma\_*$ = -0.3$\Delta\sigma\_0$ (ad-hoc ratio yielding r_* = 3.5 km; Eq. 7 – event distances sampled from the $\delta(r)$ distribution, repeated 100 times). Such distribution is not observed in Nature; (c) Same as (a) but with random uniform noise representative of spatial heterogeneities added to the regional stress field; (d) Power-law-like aftershock spatial density rho with power exponent maximum likelihood estimate q = 1.7, representative of real aftershock observations (see Fig. X2a), due to the addition of uniform noise to the static stress field.

Figure X2. Estimating the Solid Seismicity parameters from the aftershock spatial distribution: (a) Linear spatial distribution rho(r) of the four largest strike-slip mainshocks in Southern California (first aftershock generation as defined from the nearest-neighbour method). r_* = 1 km and $\delta\_+(m\_0 = 2)$ = 1.35 events/km^3 can be retrieved from the observed plateau at low r, in agreement with the SSP. Note that the spatial power-law decay at high r is similar to the one expected by the SSP in the case of a static stress field with additive uniform noise (see Fig. X1; q = 1.7 represented by the dashed black lines); (b) Aftershock productivity K for M > 6. The black line represents the Utsu law as defined by Solid Seismicity (Eq. X1, simplified case). We see that taking into account the second term of the productivity law (full second line of Eq. 12 with r_* known) has no significant impact on the result (dashed and dotted curves).

References:

Deichmann, N.: Theoretical Basis for the Observed Break in ML/Mw Scaling between Small and Large Earthquakes, Bull. Seismol. Soc. Am., 107, doi: 10.1785/0120160318, 2017.

Felzer, K. R. and Brodsky, E. E.: Decay of aftershock density with distance indicates triggering by dynamic stress, Nature, 441, 735-738, doi: 10.1038/nature04799, 2006.

Gu, C., Schumann, A. Y., Baiesi, M. and Davidsen, J.: Triggering cascades and statistical properties of aftershocks, J. Geophys. Res. Solid Earth, 118, 4278-4295, doi: 10.1002/jgrb.50306, 2013.

Hainzl, S., Brietzke, G. B. and Zöller, G.: Quantitative earthquake forecasts resulting from static stress triggering, J. Geophys. Res., 115, B11311, doi: 10.1029/2010JB007473, 2010.

King, G. C. P. and Bowman, D. D.: The evolution of regional seismicity between large earthquakes, J. Geophys. Res., 108, 2096, doi: 10.1029/2001JB000783, 2003.

Lippiello, E., de Arcangelis, J. and Godano, C.: Role of Static Stress Diffusion in the Spatiotemporal Organization of Aftershocks, Phys. Rev. Lett., 103, 038501, doi: 10.1103/PhysRevLett.103.038501, 2009.

Marsan, D. and Lengliné, O.: A new estimation of the decay of aftershock density with distance to the mainshock, J. Geophys. Res., 115, B09302, doi: 10.1029/2009JB007119, 2010.

Mignan, A., King, G. C. P. and Bowman, D.: A mathematical formulation of accelerating moment release based on the stress accumulation model, J. Geophys. Res., 112, B07308, doi: 10.1029/2006JB004671, 2007.

Mignan, A.: Retrospective on the Accelerating Seismic Release (ASR) hypothesis: Controversy and new horizons, Tectonophysics, 505, 1-16, doi: 10.1016/j.tecto.2011.03.010, 2011.

Mignan, A.: Seismicity precursors to large earthquakes unified in a stress accumulation framework, Geophys. Res. Lett., 39, L21308, doi: 10.1029/2012GL053946, 2012.

Mignan, A.: Static behaviour of induced seismicity, Nonlin. Processes Geophys., 23, 107-113, doi: 10.5194/npg-23-107-2016, 2016.

Mignan, A.: Utsu aftershock productivity law explained from geometric operations on the permanent static stress field of mainshocks, Nonlin. Processes Geophys. Discuss., doi: 10.5194/npg-2017-38, 2017.

Moradpour, J., Hainzl, S. and Davidsen, J.: Nontrivial decay of aftershock density with distance in Souther California, J. Geophys. Res. Solid Earth, 119, 5518-5535, doi: 10.1002/2014JB010940, 2014.

Richards-Dinger, K., Stein, R. S. and Toda, S.: Decay of aftershock density with distance does not indicate triggering by dynamic stress, Nature, 467, 583-586, doi: 10.1038/nature09402, 2010.

Shearer, P. M.: Space-time clustering of seismicity in California and the distance dependence of earthquake triggering, J. Geophys. Res., 117, B10306, doi: 10.1029/2012JB009471, 2012.

van der Elst, N. J. and Shaw, B. E.: Larger aftershocks happen farther away: Nonseparability of magnitude and spatial distributions of aftershocks, Geophys. Res. Lett., 42, 5771-5778, doi: 10.1002/2015GL064734, 2015.
* * *
**Fig. 1.**

[Figure]

[Figure]

**Fig. 2.**

[Figure]

---

## Author Comment (AC2) · 9 Nov 2017

Dear reviewer,

Thank you for your comments on the discussion paper by Mignan (2017). Below is my two-part answer:

1 Regarding the potential role of dynamic stress triggering

The possible contribution of dynamic triggering to aftershock productivity will be discussed in the revised manuscript: It must first be indicated that the debate around the static or dynamic origin of aftershocks has been based on the analysis of the powerlaw exponent of the spatial density of aftershocks (Felzer and Brodsky, 2006; Lipiello et al., 2009; Marsan and Lengliné, 2010; Richards-Dinger et al., 2010; Shearer, 2012; Gu et al., 2013; Moradpour et al., 2014; van der Elst and Shaw, 2015). However the original claim of a dynamic origin (Felzer and Brodsky, 2006) was later on discredited (Richards-Dinger et al., 2010) and static stress is at present the favoured theory to explain aftershock distribution in space (e.g., Moradpour et al., 2014; van der Elst and Shaw, 2015). I now also show the observed aftershock spatial distribution to support Solid Seismicity. From the SSP, and adding a uniform noise to the regional static stress field, I find a power law exponent $q = 1.7$, in agreement with the Southern California aftershock data and the literature on static stress (see my reply to reviewer #1 where I show the spatial distribution of aftershocks expected by the SSP and observed; Figs. X1d; X2a). This result will now be emphasized in both abstract and main text. Regarding the triggering of large remote events by dynamic stress (e.g., Fan and Shearer, 2016), those events have never been counted in the productivity law, declustering techniques being based on strong time-space-magnitude correlations. Even if the events shown to be triggered by dynamic stress were considered in the productivity curve, the total number of aftershocks would overshadow their role in the productivity law characteristics. Indeed, Fan and Shearer (2016) suggested the triggering of one or two M7+ aftershocks by dynamic stress per M7+ mainshock. This low number is dwarfed by the 1,000s of aftershocks produced by such mainshocks. What I infer is that static stress is sufficient to explain most of the aftershock observations over a large magnitude range, such as the aftershock spatial distribution and the aftershock productivity.

2 Regarding the geometry of the aftershock solids

The SSP expects the majority of aftershocks to occur in a volume centred on the mainshock rupture, which is clearly the case for the largest mainshocks in Southern California (Fig. 2c). This is also evident when looking at the density of aftershocks as a function of distance from rupture (new Figs. X2a – see reply to review #1). Those are "basic modern observations" that cannot be easily rejected. The result of Ross et al.

(2017) was already mentioned in the text and explained as a case in which the stress would only be partially relieved by the mainshock (line 97). Although other studies have found a deficiency of aftershocks on the main asperity, those works remain anecdotal and so cannot be considered "basic" (one M5.2 event in Ross et al.; Great 2011 Tohoku earthquake in Hasegawa et al., a giant earthquake that might show an anomalous behaviour). Figure 2c and X2a prove that it is not the case for the four major mainshocks in Southern California. Looking at smaller aftershock clusters also show no quiescence at the location of the mainshock. The red area shown in Figure 3 is also in agreement with the theory of static stress transfer (Fig. 2a-d), as described by the seminal paper of King et al. (1994). Finally, Solid Seismicity can still explain those anomalous behaviours. The aftershock deficiency case would mean that the term representative of the red volume is null, hence changing the shape of the productivity law (so the SSP is NOT "too simplified"). Unfortunately, two cases (Ross et al., 2017; Hasegawa et al., 2012) are not enough to populate such altered aftershock productivity dataset and test what modified productivity law would emerge (at least hundreds of cases would be needed). Concerning the mentioned study of van der Elst and Shaw (2015), they do not infer a deficiency of aftershocks on the mainshock fault rupture, only a deficiency in large magnitudes. This is independent of the Solid Seismicity application shown here, where only the total aftershock count is considered. In fact, van der Elst and Shaw (2015) verified that the "aftershock spatial decay is dominated by static stress transfer in the near field (several rupture lengths)" and they found $q = 1.77$ in California in good agreement with the SSP (see reply to review #1). This goes again against the dynamic stress alternative discussed in point 1.

On the last point ("further clarification regarding the time and spatial window used for aftershock counting for the case of Southern California is needed"), it will be clarified in the revised version of the manuscript that the nearest-neighbour method is used, with only first generation aftershocks considered. This will now be used systematically and figures updated accordingly, where needed.

References:

Fan, W. and Shearer, P. M.: Local near instantaneously dynamically triggered after-shocks of large earthquakes, Science, 353, 1133-1136, 2016.

Felzer, K. R. and Brodsky, E. E.: Decay of aftershock density with distance indicates triggering by dynamic stress, Nature, 441, 735-738, doi: 10.1038/nature04799, 2006.

Gu, C., Schumann, A. Y., Baisesi, M. and Davidsen, J.: Triggering cascades and sta-tistical properties of aftershocks, J. Geophys. Res. Solid Earth, 118, 4278-4295, doi: 10.1002/jgrb.50306, 2013.

Hasegawa, A. et al.: Change in stress field after the 2011 great Tohoku-Oki earth-quake, Earth Planet Sci Lett., 355-356, 231-243, 2012.

King, G. C. P., Stein, R. S. and Lin, J.: Static Stress Changes and the Triggering of Earthquakes, Bull. Seismol. Soc. Am., 84, 935-953, 1994.

Lippiello, E., de Arcangelis, J. and Godano, C.: Role of Static Stress Diffusion in the Spatiotemporal Organization of Aftershocks, Phys. Rev. Lett., 103, 038501, doi: 10.1103/PhysRevLett.103.038501, 2009.

Marsan, D. and Lengliné, O.: A new estimation of the decay of aftershock den-sity with distance to the mainshock, J. Geophys. Res., 115, B09302, doi: 10.1029/2009JB007119, 2010.

Mignan, A.: Utsu aftershock productivity law explained from geometric operations on the permanent static stress field of mainshocks, Nonlin. Processes Geophys. Discuss., doi: 10.5194/npg-2017-38, 2017.

Moradpour, J., Hainzl, S. and Davidsen, J.: Nontrivial decay of aftershock density with distance in Souther California, J. Geophys. Res. Solid Earth, 119, 5518-5535, doi: 10.1002/2014JB010940, 2014.

Richards-Dinger, K., Stein, R. S. and Toda, S.: Decay of aftershock density with

distance does not indicate triggering by dynamic stress, Nature, 467, 583-586, doi: 10.1038/nature09402, 2010.

Ross, Z. E., Kanamori, H. and Hauksson, E.: Anomalously large complete stress drop during the 2016 Mw 5.2 Borrego Springs earthquake inferred by waveform modelling and near-source aftershock deficit, Geophys. Res. Lett., 44, 5994-6001, doi: 10.1002/2017GL073338, 2017.

Shearer, P. M.: Space-time clustering of seismicity in California and the distance dependence of earthquake triggering, J. Geophys. Res., 117, B10306, doi: 10.1029/2012JB009471, 2012.

van der Elst, N. J. and Shaw, B. E.: Larger aftershocks happen farther away: Nonseparability of magnitude and spatial distributions of aftershocks, Geophys. Res. Lett., 42, 5771-5778, doi: 10.1002/2015GL064734, 2015.

---

## Referee Comment (RC2) · Anonymous Referee #2 · 20 Nov 2017

This is an interesting paper which correlates the Utsu aftershock productivity with the geometric operations on the permanent static stress field. The paper is well written and I have very minor comments on the manuscript as indicated below.

1. For Figure 2, several hours after the 1992 Landers earthquake, the largest aftershock (or triggered earthquake), Big Bear earthquake, occurred southwest of the mainshock source region. I think it's better to mention in the text (around Lines 90) that these off-fault triggered seismicity also happened due to static stress changes imparted by the mainshock, while these triggered seismicity are out of topic in this paper. (If my understanding is correct, please neglect if I'm wrong) 2. In Figure 2a, the author

assumed the regional stress of 10 bar. But, I think that this assumed regional stress is too small to cause earthquakes, because a stress drop basically ranges 10-100 bars (Kanamori and Anderson). Furthermore, I think that it is not so obvious whether on-fault aftershocks are due to static stress changes imparted by the mainshock or not. It's better to mention this point more carefully by referring several previous studies.

---

## Author Comment (AC3) · 27 Nov 2017

Dear reviewer,

Thank you for your comments on the discussion paper by Mignan (2017).

As per your suggestion, I will now mention the case of triggered off-fault seismicity, as exemplified by the Big Bear earthquake, which is indeed "also due to static stress changes imparted by the mainshock". The anisotropic effects observed on nearby faults can be explained by the Solid Seismicity Postulate, as shown already in Figure 5 of Mignan (2016). This will now be explained in the text. Since such heterogeneities

in space are not systematic, they are indeed "out of topic in this paper", which is concerned with the general productivity law that applies to all mainshocks on average.

Regarding Figure 2a, 10-bars seems like a reasonable value for a stress drop. Looking at Figure 5 of Abercrombie and Leary (1993), observations are centred on 1-100 bar in log10 scale. Then Figure 2a represents the case where the stress drop counterbalances the regional deviatoric stress, so whatever value is used, the final outcome would be the same (Figures 2a and 2d being similar to Figure 3 of King et al., 1994). Finally, a reference to Miller et al. (2004) will be added to indicate that additional physical processes (such as trapped high pressure gas) may also explain part of the on-fault aftershock activity.

References:

Abercrombie, R. and Leary, P.: Source parameters of small earthquakes recorded at 2.5 km depth, Cajon Pass, Southern California: Implications for earthquake scaling, Geophys. Res. Lett., 20, 1511-1514, 1993.

King, G. C. P., Stein, R. S. and Lin, J.: Static Stress Changes and the Triggering of Earthquakes, Bull. Seismol. Soc. Am., 84, 935-953, 1994.

Mignan, A.: Static behaviour of induced seismicity, Nonlin. Processes Geophys., 23, 107-113, doi: 10.5194/npg-23-107-2016, 2016.

Mignan, A.: Utsu aftershock productivity law explained from geometric operations on the permanent static stress field of mainshocks, Nonlin. Processes Geophys. Discuss., doi: 10.5194/npg-2017-38, 2017.

Miller, S. A., Collettini, C., Chiaraluce, L., Cocco, M., Barchi, M. and Kaus, B. J. P.: Aftershocks driven by a high-pressure CO2 source at depth, Nature, 427, 724-727

---

## Author Response (AR1)

Dr. Arnaud Mignan

Institute of Geophysics,

Swiss Federal Institute of Technology, Zürich

NO H66, Sonneggstrasse 5

CH-8092 Zürich arnaud.mignan@sed.ethz.ch

December 2017

Dear Editor and Reviewers,

Please find below my answers to your comments, as given in the discussion section of NPG. I now additionally refer to line numbers of the annotated version of the modified manuscript to describe specific changes. Those additions are highlighted in blue in the present reply. The main changes made to the article are the addition of two new subsections (2.2 and 3.2 on the aftershock spatial distribution) and of two new figures (Figs. 4-5).

Sincerely,

Arnaud Mignan

**Anonymous Referee #1**

The MS presented by Dr. Mignan intends to provide the background of the aftershock productivity law where the number of aftershock is proportional to the exponential of the magnitude (M) of a mainshock. On the basis of "Solid Seismicity Postulate"(SSP), the author derives the formula of the expected number of aftershocks as a function of M which agrees with the productivity law originally suggested by Utsu [1970]. The derived formula has a break in the log-linear relationship between the aftershock productivity and M whereas the break is not found through the analysis of real aftershock data. The author suggests that this inconsistency is caused by an aftershock selection bias with a numerical simulation.

I have two major concerns on this MS as shown below. a) I do not understand well what new significant results are in this MS. In Hainzl et al [2010, JGR], the aftershock productivity law has already been reproduced with a numerical simulation. The simulation is based on the "clock-advanced" model, which is a simple but realistic physical assumption. By contrast, SSP is too simple, and because of this simplification its physical background seems obscure and unrealistic. Furthermore, the postulate has not been supported by real data (In some of the author's previous papers, seismicity model derived from SSP has been applied to real seismicity data.

Note that, however, only temporal patterns of seismic activity are analyzed. To validate SSP where we have only three seismicity levels in space, it is indispensable to reproduce spatial patterns of real earthquakes.). This MS does not show any convincing motivation to explain the productivity law with such an unsupported postulate. I understand that sometimes it is important to introduce a (too) simple model/assumption for explaining an empirical law. However, it is also important to provide some new and meaningful perspective as a result of the introduction. The results shown in this MS do not go beyond the results of Hainzl et al. [2010], and therefore the introduction of SSP is unproductive.

b) In the end of Section 3, the author suggests the break in scaling in the after- shock productivity data (Eq.(16)). However, as a result of the analysis of the real aftershock data, no break is found (L.182-183). To explain the result of "no break", in Section 4 the numerical simulation with the ETAS model was conducted. Then, the author ascribes this result to the "aftershock selection bias" (L.206-207) in the numerical simulation. The author's conclusion is one possibility, but it is also possible that Eq.(16) is incorrect; the numerical simulation shown in Section 4 is inconclusive, and I do not understand what is the meaning of showing such a vague consequence. The application of an aftershock selection approach having a serious problem (the bias in this case) itself is inappropriate. Why does the author use any other approach which does not contain such a problem? In other words, only a negative possibility for the postulate is shown and no positive support is not given in the present form of this MS. To my opinion, this is another major drawback of this MS.

Some further comments:

1) Introduction of the Zero-Inflated Poisson (ZIP) distribution The reason of the introduction of the ZIP distribution is described in L.165-166 ("this approach ... zero aftershock"), but this explanation seems insufficient. Behind the ZIP distribution, we have the following assumption. We have two possibilities: the first is that the number of events follows a Poisson distribution, and the second is that it is deterministically equal to zero. One of these two possibilities is chosen through the Bernoulli distribution. As far as I know, a physical (seismological) process corresponding to the Bernoulli distribution is unclear in generating earthquakes. If the author persists in introducing the ZIP distribution, explain what is the physical process.

2) The simulation shown in Section 4  This simulation is based on the ETAS model, and this violates the self-consistency of this MS. As seen in $g(x,y|M)$ of Eq.(17), the spatial density of aftershocks gradually decays with the distance from a parent event. This property completely disagree with SSP (see Eq.(5)). For the self-consistency, the simulated spatial (and temporal) pattern of earthquakes should be generated on the basis of SSP.

3) α = 2.04 (L.197) I do not understand how the author incorporated this information (value) into Eq.(16).

**Reply to Anonymous Referee #1**

Dear reviewer,

Thank you for your comments on the discussion paper by Mignan (2017). Below is my two-part answer to (1) show that the Solid Seismicity Postulate is supported by seismicity data and (2) discuss in more detail the mismatch between theoretical scaling break and lack of break in real data. A third section answers to your other comments.

**1 Support of the Solid Seismicity Postulate (SSP) by aftershock data**

The SSP should indeed be verified to be consistent with the spatial distribution of seismicity data (see new results in abstract lines 16-20). I first clarify that the step-like function of event density in space is only expected for the case of an idealised smooth static stress field (lines 163-165). I now compare this case (new Fig. 4a-b) with the case of a stress field with uniform noise (Fig. 4c-d). While the ideal case is used to develop analytical solutions, a heterogeneous stress field described by additive uniform noise was already used in past studies to simulate non-stationary background events (King and Bowman, 2003; Mignan et al., 2007; Mignan, 2011). Addition of such noise blurs the "aftershock solid", which reflects in the aftershock spatial density distribution, switching from a step function to a power-law of the form $\rho(r) \propto r^{-q}$, with ρ the linear spatial density and exponent $q = 1.96$ (the 1.7 value given in the discussion post was erroneous, as I had used the wrong MLE formulation – both values remain within the $q$-range given in the literature. q = 1.96 better fits the tail of the power-law as shown in Fig. 4d). Figure 4 was inserted in the revised manuscript and a new subsection added, titled "2.2. Validation of the Solid Seismicity Postulate" (lines 162-189).

As shown in Figure 5 in the revised manuscript, the power law exponent obtained from the SSP with noisy static stress field matches the power law exponent found in Southern California. In the literature, $1.3 < q < 2.5$ was found for California (Felzer and Brodsky, 2006; Lipiello et al., 2009; Marsan and Lengliné, 2010; Richards-Dinger et al., 2010; Shearer, 2012; Gu et al., 2013; Moradpour et al., 2014; van der Elst and Shaw, 2015). This demonstrates that the SSP is not "*too simple*" or "*unrealistic*". Comparison of Figure 4d with Figure 5a shows that "*the spatial patterns of real earthquakes are reproduced*" by the SSP (i.e., the power-law behaviour) with a realistic $q$-value (without any tuning required). A short review of past studies on the spatial distribution of aftershocks is now given lines 168-174 and a discussion of Figure 5 added in the new section 3.2 "Aftershock spatial density distribution" (lines 202-236) of the revised manuscript (section 3 "Observations & Model Fitting" being now separated in 3 subsections).

This work goes beyond the results of Hainzl et al. (2010) since an analytical formulation is explicit while the physical driver of a simulation output is implicit and potentially ambiguous (see new results in abstract lines 26-28). In the King and Bowman (2003) study for example, a power-law behaviour of precursory seismicity emerged from their static stress simulations. However the result was ambiguous. It was not clear if the behaviour emerged from the stress field geometry, implemented Gutenberg-Richter power-law, or else. It led to the first study on Solid Seismicity, which demonstrated that the power-law time-to-failure equation derived from the geometry of the stress field (Mignan et al., 2007). While such ambiguity may not be present in the simulations of Hainzl et al. (2010), we are still left wondering which parameters are critical to the emergence of the Utsu productivity law, i.e., "*it remains unclear how $K_0$ and $\alpha$ relate to the underlying physical parameters*" (line 50).

Here are two "*new and meaningful perspectives as a result of the introduction*" of the SSP (new section 3.2 and extended section 3.3, new figure 5):
(*i*) It is first of importance to demonstrate that the Solid Seismicity theory can explain the aftershock productivity law, since it already explains both tectonic foreshocks (Mignan, 2012) and induced seismicity (Mignan, 2016). If such physical framework can explain the main seismicity patterns observed in Nature, it becomes a potential candidate for a unified theory of seismicity.
(*ii*) Figure 5 (and the new section 3.2 and extended section 3.3) goes farer into the Solid Seismicity analysis, showing how to estimate its main parameters (intermediary parameter $r_*$, main parameters $\delta_+$ and $\Delta\sigma_*$). We first note that the $q = 1.96$ theoretical estimate (SSP + uniform noise) is compatible with observations (Fig. 5a). I here focus on the largest mainshocks to avoid the scattering and scaling break issues at small $M$. On the same plot, we can roughly estimate $r_* = 1$ km (maximum $r$ at which the $\rho$ plateau breaks – in analogy with Fig. 4d). It is constant for any large $M$ ($> M_{break}$) since the stress drop is a constant, $c = w_0$ is a constant, and $\Delta\sigma_*$ is also *a priori* a constant (one of the 2 main parameters of the Solid Seismicity approach; Eq. 7). See lines 215-218 (section 3.2). Now let us calculate $\delta_+$ from the commonly used parameter $K_0$ (section 3.3 lines 238-301). We first note from Eq. (11) that the second term is negligible for large M, yielding

$$K(M > M_{break}) \approx 2\delta_+(m_0)r_*\exp[ln(10)(M - 4)] \qquad \text{(X1 – new 18)}$$

Rearranging $m_0$ and $M$-4 and comparing to the original Utsu Eq. (1), we get

$$\delta_+(m_0) = \frac{K_0\exp[ln(10)(4-m_0)]}{2r_*} \qquad \text{(X2 – new 19)}$$

With $\alpha = \ln(10)$ fixed and $K_0$ estimated from the MLE for $M > 6$, we get $K_0 = 0.025$ and thus $\delta_+(m_0 = 2) = 1.23$ events/km$^3$ (fit represented in Fig. 5b). If correct, the linear density below $r_*$ (plateau) for any given large $M$ should be

$$\rho(r < r_*, M) = \delta_+ \exp[\ln(10)(M - 4)] \qquad\qquad \text{(X3 – new 14)}$$

which is represented on Fig. 5a and matches the data (Eq. (14) simply calculates the linear density of events $\rho$ from the volumetric density of events $\delta_+$) (lines 222-228). This suggests that $\delta_+$ is also constant, at least for the four largest strike-slip mainshocks in Southern California (line 297). One could have also estimated $\delta_+$ directly from $\rho(r)$ (as done for $r_*$) to directly derive the aftershock productivity law of Southern California with Eq. (18). This shows the direct link between aftershock productivity and aftershock spatial distribution (or geometry). As for the parameter $\Delta\sigma_*$, its estimation remains ambiguous as it depends on the seismogenic width $w_0$. We get the ratio $\Delta\sigma_*/\Delta\sigma_0 = \{-0.5, -1.0, -1.4\}$ for $w_0 = \{5, 10, 15\}$ km, respectively (Eq. 7) (lines 218-221).

This of course remains a preliminary analysis. However I hope that additional analyses of aftershocks, foreshocks as well as induced seismicity in different regions will provide useful information as to the distribution of the $\Delta\sigma_*$ and $\delta_+$ parameters. Are they universal? Is a same regional value applicable to all types of seismicity? Are there any correlations? Those are important questions I wish to answer in the near future. To do so, the theoretical framework must first be conveyed for each class of seismicity pattern. See lines 363-368 in the conclusion.

**2 Theoretical scaling break & mismatch with seismicity data**

The discussion paper already indicates that: "*Possible biases of aftershock selection may explain the lack of break*" (lines 18-20, abstract) and "*while such a bias is possible, it yet does not prove that the break in scaling exists*" (line 208) – This clearly suggests that it is only one possible option. It is indeed a weak argument (since based on a negative result) but it is so far the best one available (all existing declustering techniques assuming no break in magnitude). "*It is also possible that Eq. (16) is incorrect*", true, but so would the clock-advance model in such premise, which the reviewer describes as "*a simple but realistic physical assumption*". No explanation for the lack of break in real data was given in Hainzl et al. (2010). The present paper provides one possible explanation. Any criticism on the scaling break mismatch shall apply the same way to the present study and the published one of Hainzl et al. (2010). An alternative view is that both studies found the same scaling break, hence supporting this result as characteristic of the static stress process.

Following on the new results presented in Figure 4d, the explanation of lack of break due to aftershock selection bias becomes a more realistic one. It is NOT "*a vague consequence*" since any study of the aftershock productivity law is based on the use of such a declustering method. The ETAS simulation does NOT "*violate the self-consistency of this MS*" since the power-law spatial distribution is now shown to be verified by the SSP (line 321). The theoretical value $q = 1.96$ is close to the value I already used in the ETAS simulations ($q = 1.47$) and observed here for the largest strike-slip mainshocks (Fig. 5a). Since the aftershock selection bias is only one option, another alternative is now discussed: The proposed productivity equation assumes moment magnitude while the earthquake catalogue is in local magnitude. Deichmann (2017) recently demonstrated that while $M_L \propto M_w$ at large $M$, $M_L \propto 1.5 M_w$ at small $M$. This would cancel the kink observed in the real data. However the scaling break predicted by Deichmann (2017) occurs at several magnitude units below the geometric one expected by static stress (new lines 348-354).

**3 Other aspects**

On the introduction of the Zero-Inflated Poisson (ZIP) distribution: Explaining the distribution of earthquakes, from the static stress process to their occurrence on a fractal network of faults remains out of the scope of the present study. Since the ZIP does not lead to significant changes in the $\alpha$-value and since section 3 is now completed with an analysis of the spatial distribution of aftershocks, the ZIP part has been deleted from the revised manuscript.

On $\alpha = 2.04$ (line 197): This is the maximum likelihood estimate of $\alpha$ obtained for Southern California in the present study (see line 164). $\alpha$ is thus constrained from large magnitude data (Fig. 4a) and the simulated break at lower magnitudes is estimated from the theoretical value $3/2\ \alpha$. Values of $\alpha$ are now given for different magnitude M ranges and explained (lines 248-250, 309-313, 334).

**Anonymous Referee #2**

This is an interesting paper which correlates the Utsu aftershock productivity with the geometric operations on the permanent static stress field. The paper is well written and I have very minor comments on the manuscript as indicated below.
1. For Figure 2, several hours after the 1992 Landers earthquake, the largest aftershock (or triggered earthquake), Big Bear earthquake, occurred southwest of the mainshock source region. I think it's better to mention in the text (around Lines 90) that these off-fault triggered seismicity also happened due to static stress changes imparted by the mainshock, while these triggered seismicity are out of topic in this paper. (If my understanding is correct, please neglect if I'm wrong) 2. In Figure 2a, the author assumed the regional stress of 10 bar. But, I think that this assumed regional stress is too small to cause earthquakes, because a stress drop basically ranges 10-100 bars (Kanamori and Anderson). Furthermore, I think that it is not so obvious whether on- fault aftershocks are due to static stress changes imparted by the mainshock or not. It's better to mention this point more carefully by referring several previous studies.

**Reply to Anonymous Referee #2**

Dear reviewer,

Thank you for your comments on the discussion paper by Mignan (2017).

As per your suggestion, I now mention the case of triggered off-fault seismicity, as exemplified by the Big Bear earthquake, which is indeed "also due to static stress changes imparted by the mainshock". The anisotropic effects observed on nearby faults can be explained by the Solid Seismicity Postulate, as shown already in Figure 5 of Mignan (2016). This is now explained in the text. Since such heterogeneities in space are not systematic, they are indeed "out of topic in this paper", which is concerned with the general productivity law that applies to all mainshocks on average. See new lines 229-236 and new curve in Figure 5a.

Regarding Figure 2a, 10-bars seems like a reasonable value for a stress drop. Looking at Figure 5 of Abercrombie and Leary (1993), observations are centred on 1-100 bar in log10 scale. Then Figure 2a represents the case where the stress drop counterbalances the regional deviatoric stress, so whatever value is used, the final outcome would be the same (Figures 2a and 2d being similar to Figure 3 of King et al., 1994). Finally, a reference to Miller et al. (2004) has been added to indicate that additional physical processes (such as trapped high pressure gas) may also explain part of the on-fault aftershock activity (line 384).

**Referee N. Wetzler**

The manuscript examines the empirical relationship of the power law aftershock productivity law. The author introduces (not only in this study) the Solid Seismicity Postulate (SSP) to predict the first order mainshock's geometrical static stress perturbation on the crustal ambient stress. The model defines two basic ruptures with respect to the free surface predicting a magnitude dependent deficiency when the rupture hits the surface. Using this physical model he explains the empirical observation. The manuscript is written well and figures are useful. I have two general comments: 1) The role of dynamic triggering. In general aftershock productivity is a product of the static and dynamic perturbation superimposed on the regional seismic susceptibility, or faults state [Dieterich, 1994]. Many examples for both dynamic triggering and Coulomb stress explain aftershocks occurrence. Due to the rapid decay of the static stress field, cases of "pure" dynamic triggering are common beyond several fault dimensions rom the mainshock [e.g. Fan and Shearer, 2016]. In the periphery of the fault, the Coulomb stress field and dynamic stress field overlap with a similar fashion, and it is unclear how they interact. My main concern is that the author does not discuss the contribution of dynamic triggering to the aftershock productivity. Does the fact that the predicted "kink" in the aftershock productivity from the geometrical interaction with the surface is due to enhancement of the dynamic triggering? 2) The geometry of the SSP. The first order shape of the SSP is not obvious to me. The geometry of the induced area is predicting a volumetric increase in static stress changes along the rupture area (red in Figure. 3). The rupture of faulted area is the expression of the coseismic slip responding to the elastic rebound. This predicts different degrees of relaxation with respect to the main- shock magnitude and the occurrence of the event in the seismic cycle. In the case of "complete" stress drop the rupture area is predicted to present spatial deficiency in productivity and some variations in the field with respect to the fault complexity. Several papers demonstrate the deficiency in aftershocks at the asperity with the majority of the seismicity focused on the periphery of the fault [Hasegawa et al., 2012; van der Elst and Shaw, 2015; Ross et al., 2017] represented by the orange volume in Figure. 3. My concern is that this model (SSP) is too simplified and does not incorporate basic modern observations.

3) Further clarification regarding the time and spatial windows used for aftershock counting for the case of Southern California is needed

Dieterich, J. H. (1994), A constitutive law for rate of earthquake production and its application to earthquake clustering, J. Geophys. Res., 99(B2), 2601–2618, doi:10.1029/93JB02581. van der Elst, N. J., and B. E. Shaw (2015), Larger aftershocks happen farther away: Nonseparability of magnitude and spatial distributions of aftershocks, Geophys. Res. Lett., 42(14), 5771–5778, doi:10.1002/2015GL064734. Fan, W., and P. M. Shearer (2016), Local near instantaneously dynamically triggered aftershocks of large earthquakes, Science (80-. )., 353(6304), 1133–1136, doi:10.1126/science.aag0013. Hasegawa, A., K. Yoshida, Y. Asano, T. Okada, T. Iinuma, and Y. Ito (2012), Change in stress field after the 2011 great Tohoku-Oki earth- quake, Earth Planet. Sci. Lett., 355–356, 231–243, doi:10.1016/j.epsl.2012.08.042. Ross, Z. E., H. Kanamori, and E. Hauksson (2017), Anomalously large complete stress drop during the 2016 M w 5.2 Borrego Springs earthquake inferred by wave- form modeling and near-source aftershock deficit, Geophys. Res. Lett., 1–8, doi:10.1002/2017GL073338.

**Reply to Referee N. Wetzler**

Dear reviewer,

Thank you for your comments on the discussion paper by Mignan (2017). Below is my two-part answer:

**1 Regarding the potential role of dynamic stress triggering**

The possible contribution of dynamic triggering to aftershock productivity is now discussed in the revised manuscript (lines 171-177):
It must first be indicated that the debate around the static or dynamic origin of aftershocks has been based on the analysis of the power-law exponent of the spatial density of aftershocks (Felzer and Brodsky, 2006; Lipiello et al., 2009; Marsan and Lengliné, 2010; Richards-Dinger et al., 2010; Shearer, 2012; Gu et al., 2013; Moradpour et al., 2014; van der Elst and Shaw, 2015). However the original claim of a dynamic origin (Felzer and Brodsky, 2006) was later on discredited (Richards-Dinger et al., 2010) and static stress is at present the favoured theory to explain aftershock distribution in space (e.g., Moradpour et al., 2014; van der Elst and Shaw, 2015) (lines 171-177 of the new section 2.2).
I now also show the observed aftershock spatial distribution to support Solid Seismicity. From the SSP, and adding a uniform noise to the regional static stress field, I find a power law exponent $q = 1.96$, in agreement with the Southern California aftershock data and the literature on static stress (see my reply to reviewer #1 where I show the spatial distribution of aftershocks expected by the SSP and observed; Figs. 4d; 5a). This result is now be emphasized in both abstract (lines 16-20) and main text (new sections 2.2 and 3.2, lines 321, 371-373).
Regarding the triggering of large remote events by dynamic stress (e.g., Fan and Shearer, 2016), those events have never been counted in the productivity law, declustering techniques being based on strong time-space-magnitude correlations. Even if the events shown to be triggered by dynamic stress were considered in the productivity curve, the total number of aftershocks would overshadow their role in the productivity law characteristics. Indeed, Fan and Shearer (2016) suggested the triggering of one or two M7+ aftershocks by dynamic stress per M7+ mainshock. This low number is dwarfed by the 1,000s of aftershocks produced by such mainshocks (lines 174-177).
What I infer is that static stress is sufficient to explain most of the aftershock observations over a large magnitude range, such as the aftershock spatial distribution and the aftershock productivity.

**2 Regarding the geometry of the aftershock solids**

The SSP expects the majority of aftershocks to occur in a volume centred on the mainshock rupture, which is clearly the case for the largest mainshocks in Southern California (Fig. 2c). This is also evident when looking at the density of aftershocks as a function of distance from rupture (new Fig. 5a – see reply to review #1). Those are "*basic modern observations*" that cannot be easily rejected.

The result of Ross et al. (2017) was already mentioned in the text and explained as a case in which the stress would only be partially relieved by the mainshock (line 97). Although other studies have found a deficiency of aftershocks on the main asperity, those works remain anecdotal and so cannot be considered "basic" (one M5.2 event in Ross et al.; Great 2011 Tohoku earthquake in Hasegawa et al., a giant earthquake that might show an anomalous behaviour). Figure 2c and 5a prove that it is not the case for the four major mainshocks in Southern California. Looking at smaller aftershock clusters also show no quiescence at the location of the mainshock. The red area shown in Figure 3 is also in agreement with the theory of static stress transfer (Fig. 2a-d), as described by the seminal paper of King et al. (1994). Finally, Solid Seismicity can still explain those anomalous behaviours. The aftershock deficiency case would mean that the term representative of the red volume is null, hence changing the shape of the productivity law (so the SSP is NOT "*too simplified*"). Unfortunately, two cases (Ross et al., 2017; Hasegawa et al., 2012) are not enough to populate such altered aftershock productivity dataset and test what modified productivity law would emerge (at least hundreds of cases would be needed) (lines 174-177).

Concerning the mentioned study of van der Elst and Shaw (2015), they do not infer a deficiency of aftershocks on the mainshock fault rupture, only a deficiency in large magnitudes. This is independent of the Solid Seismicity application shown here, where only the total aftershock count is considered. In fact, van der Elst and Shaw (2015) verified that the "*aftershock spatial decay is dominated by static stress transfer in the near field (several rupture lengths)*" and they found $q = 1.77$ in California in good agreement with the SSP (see reply to review #1). This goes again against the dynamic stress alternative discussed in point 1. Reference to van der Elst and Shaw (2015) has been added.

On the last point ("*further clarification regarding the time and spatial window used for aftershock counting for the case of Southern California is needed*"), it is now clarified in the revised version of the manuscript that the nearest-neighbour method is used, with only first generation aftershocks considered. This is now used systematically and figures have been updated accordingly, where needed (lines 197, 229-233, Figs. 5, 6).

[revised manuscript text omitted]

Arnaud Mignan 29.11.2017 16:09

Arnaud Mignan 4.12.2017 15:05

Arnaud Mignan 4.12.2017 15:05

Arnaud Mignan 4.12.2017 15:05

Arnaud Mignan 5.12.2017 10:37

---

## Author Response (AR2)

Dr. Arnaud Mignan

Institute of Geophysics,

Swiss Federal Institute of Technology, Zürich

NO H66, Sonneggstrasse 5

CH-8092 Zürich arnaud.mignan@sed.ethz.ch

February 2018

Dear Editor Ilya Zaliapin,

Please find below my answers to your comments, highlighted in blue. I agree that the Solid Seismicity Postulate remains to be verified, so the text was changed to clarify this important point. The only point where I disagree is to discard the term "theory". As I explain below, what is proposed can be defined as a theory. I hope that this reply answers to all of your remaining concerns.

Sincerely,

Arnaud Mignan

**Editor's comments**

Comments to the Author:
The revised paper shows a significant improvement in terms of clarity and justification of the main statements. I find that most of the technical comments raised by the reviewers were addressed in this revision. At the same time, there remain several conceptual issues that are being debated by the author. Resolving these issues (mainly, by revising the current text and conclusions) would make the paper acceptable to publication in NPG.

One of the main concerns is that the SSP, the main methodological tool of the work, has not been shown to be a physically justified principle that drives the observed seismicity. Specifically, the SSP seems to be a practical toy model that can be appropriately ramified (e.g., by adding noise, like in Fig. 4) to look consistent with the data. This might be not surprising though: the activity (suitably defined) of aftershocks generally decays away from a mainshock, so a model formulated in terms of decay (continuous or step-like) could be made consistent with data. This observation alone is insufficient to prove the validity of SSP. Nevertheless, the paper claims that the SSP has been validated (l. 16) and the SSP is a "proper approach" (l. 367) that can explain "most empirical laws observed in seismicity" (l. 356). These claims are unsupported by the analysis presented in the work; I think such claims can distract a reader and harm a potential impact of the work. A possible resolution would be to explicitly introduce SSP as an assumption and illustrate how it can be used to make inference regarding the productivity law. Such analysis might be interesting from various points of view and can stimulate further research. The work nicely illustrates how a basic assumption can be transformed into testable statements regarding the main laws of seismicity; however an attempt to claim that such an assumption is an actual law of seismicity might be premature.

I consider the suggested revisions as minor. However, if the author insists on physical validity of the SSP as a new paradigm in understanding seismicity, a substantial further research and justification will be required.

*I modified the text accordingly. The term "validated" was replaced everywhere by "tested" (lines 17, 106, 186). I also changed "suggests that the SSP is a proper approach" to "shows that the SSP is consistent with large aftershock observations once uniform noise is added to the stress field" (line 361) and that other types of noise have yet to be tested (line 362). It was already indicated in the same paragraph that the SSP remains to be proven and is "so far a rather convenient and pragmatic assumption" (line 358). I added that "This result alone is however insufficient to prove the validity of the SSP" (line 217) and finally deleted the sentence: "most empirical laws observed in seismicity populations can be explained by…".*

I list other comments below:

It is worth adding a brief summary and discussion of the findings by Hainzl et al. (2010). Their results seem important for understanding the motivation and some of the results of this work.

*I added a brief description of the Hainzl et al. (2010) approach and interpretation lines 175-180.*

l. 11: parameter K needs a better definition, e.g. "K is the number of aftershocks triggered by a given mainshock of magnitude M "

*done.*

l. 14: "Solid Seismicity Postulate" (please insert Postulate)

*done.*

ll. 25-28: Please rewrite the sentence, possibly splitting it into two: one regarding the estimations and the other on the necessity to prove the existence of the kink.

*I split the sentence in two, now clearly separating the part on the kink and the part on parameter estimation.*

l. 31: explain "most robust"

*I replaced "most robust" by "one of the most studied"*

ll. 32-33: explain what empirical laws are mentioned here

I added "*such as the Modified Omori Law*" for the temporal one. However there is no name available for the spatial law. The productivity law is defined as Utsu law in the next sentence.

l. 45, Eq(2): Define N and A

done.

ll. 65: "Solid Seismicity Postulate" (add Postulate)

done.

l. 71: Revise the subsection title (sounds vague at the moment)

Now changed to "Demonstration of the productivity law by geometric operations"

l. 72: "a geometrical theory of seismicity" does not seem justified. Is it possible to merely formulate the postulate, without calling it a "theory"?

I would prefer to keep the term "theory". A theory is an explanation that can be repeatedly tested, which is here possible as all parameters are clearly described in algebraic equations. Each seismicity patterns is explicitly categorized into background, quiescence and activation based on a spatial event density definition. On Wikipedia, we read "the strength of a scientific theory is related to the diversity of phenomena it can explain and its simplicity". The proposed theory can describe aftershock productivity, foreshocks (GRL2012) and induced seismicity (NPG2016) solely based on 2 parameters. It has yet to be fully tested to become an established theory or to be rejected, but it is a theory nonetheless. The idea behind the SSP is new and cannot be related to any existing theory of seismicity. The introduced parameters are also new and not related to any other existing seismicity framework. It also represents an abstract concept that generalizes the definition of seismicity patterns in space and time. Solid Seismicity is also not a model but different models can be developed from it, eg an aftershock production model (this paper), a precursory seismicity model or an induced seismicity model (previous papers). I hope that you can agree with this definition.

l. 75: "strictly categorized" needs to be explained

Now defined lines 96-98 as a "sort of hard labelling". Any seismicity population is either in one of the 3 classes defined in the SSP and no other.

l. 79, Eq. (5): Please define sigma and delta (explanation + units), and "background stress amplitude range".

done.

l. 92: Please define r and explain the equation.

done.

l. 113, Eq. (7): the comma must appear after the Eq. in line 113, not in line 114.

corrected.

l. 120: "rupture surface area" (add area)

added.

l. 163: Please explain "step-like spatial behavior". Does this refer to the spatial density of aftershocks?

this is correct, now clarified.

l. 173: "discredited" in this context sounds as too strong of a term. Can you revise?

I changed "*discredited*" to "*questioned*".

l. 244: Why "Poisson process"?

Now explained, as "*representing the stochasticity of the count K of aftershocks produced by a mainshock at any given time*."

l. 351: Please justify "physical". Eq. (12) is a consequence of an ad-hoc SS postulate; its connection to physical principles has not been established.

I removed "*physical*". It was meant to refer to parameters based on physical properties, such as stress or event count.

l. 355-357: I do not find this conclusion justified by the presented analysis (see above).

This sentence has been removed.

ll. 365-367: The ability to reproduce the scaling parameter q should be critically assessed against the number of assumptions and parameters involved in this estimation.

I clarified that $q$ was retrieved once a uniform noise was added to the stress field and that "*the impact of other types of noise on q has yet to be investigated*" (lines 362-371)

Throughout the paper:
Please avoid using consecutive parentheses, like in "(Kanamori and Anderson, 1975) (Fig. 1d)."
Please check punctuation marks (commas, periods) in equations.

Done.

[revised manuscript text omitted]

---

## Author Response (AR3)

Dr. Arnaud Mignan

Institute of Geophysics,

Swiss Federal Institute of Technology, Zürich

NO H66, Sonneggstrasse 5

CH-8092 Zürich arnaud.mignan@sed.ethz.ch

February 2018

Dear Editor Ilya Zaliapin,

Please find below my answer to your latest comment. I hope that by the few minor changes made, I correctly addressed the raised issue.

Sincerely,

Arnaud Mignan

**Editor's comments**

Comments to the Author:
I would like to thank the author for the next round of revisions, which further improved the readability of the paper. The revised version reveals a methodological issue that needs to be clarified.

It is stated in ll. 71-72 (marked version of the paper) that "The aim of the present article is to explain the Utsu aftershock productivity equation". At the moment is it not clear, however, how the derivations of Section 2.1 can explain the Utsu law and why the SSP is important here. Specifically, my understanding is that Eq. (12) is the final suggested explanation for the Utsu law. Furthermore, only a part of this equation — I mean S(M) in some power — is used to explain the Utsu law. I list here the assumptions used to derive Eq. (12): (i) rupture surface area S(M) scales with event magnitude, (ii) there exist a connected region around the mainshock rupture that accommodates the aftershocks (aftershock solid), and (iii) large EQs rupture the seismogenic layer, while small ones develop in a volume. The key equations that gives the sought result are: r(M) ~ S(M)^alpha (alpha=1/2 or 0 depending on the mainshock size) and V(M) ~ r(M)S(M).

If my understanding is correct, the role of the SSP remains unclear. One can make multiple alternative assumptions regarding the intensity of events, and arrive at a similar scaling, which follows from straightforward geometric considerations and scaling of the surface area with magnitude. In other words, the presented derivations show that the SPP does not contradict the Utsu law, and can suggest a particular parameterization for the law's constants. This is different from "explaining" the law. Accordingly, it is important to justify the necessity of SPP in the presented derivations and separate the routine calculation of parameters (constants) from the derivation of the main scaling part of the equation.

The above issue can be readily addressed by removing the claims that SPP explains the Utsu law, and presenting SPP as a particular parameterization for this and other empirical regularities.

I modified the text accordingly, as follows:

Line 14: "We explain this law based on the SSP" changed to "We parameterize this law using the SSP"
Lines 63-69: "describe" instead of "explain", "where the Eq. (4) scaling is parameterized using the SSP" added.
Line 342: "describe" instead of "explain".

I however keep the term "explain" when referring to other empirical laws (foreshocks and induced seismicity) as the functional forms are directly derived from the SSP. I agree that it was not the case for aftershocks. While I provided a new formula for the aftershock production, the Utsu scaling only emerges when injecting eq. 4. This is now clarified.

Note that I also changed the specific units given in the SSP definition to "stress unit" and "number of events per unit of volume" to remain generic (lines 81-84).

[revised manuscript text omitted]